# Molecular basis of ligand-dependent Nurr1-RXRα activation

Xiaoyu Yu[1,2†], Jinsai Shang[2‡], Douglas J Kojetin[2,3*§, #†]

[1]Skaggs Graduate School of Chemical and Biological Sciences at Scripps Research, Jupiter, United States; [2]Department of Integrative Structural and Computational Biology, Scripps Research and UF Scripps Biomedical Research, Jupiter, United States; [3]Department of Molecular Medicine, Scripps Research and UF Scripps Biomedical Research, Jupiter, United States

*For correspondence: douglas.kojetin@vanderbilt.edu

Present address: †Department of Biochemistry, Vanderbilt University School of Medicine, Nashville, United States; ‡School of Basic Medical Sciences, Guangzhou Laboratory, Guangzhou Medical University, Guangzhou, China; §Center for Structural Biology, Vanderbilt University School of Medicine, Nashville, United States; #Vanderbilt Institute of Chemical Biology, Vanderbilt University School of Medicine, Nashville, United States

Competing interest: The authors declare that no competing interests exist.

## Abstract

Small molecule compounds that activate transcription of Nurr1-retinoid X receptor alpha (RXRα) (NR4A2-NR2B1) nuclear receptor heterodimers are implicated in the treatment of neurodegenerative disorders, but function through poorly understood mechanisms. Here, we show that RXRα ligands activate Nurr1-RXRα through a mechanism that involves ligand-binding domain (LBD) heterodimer protein-protein interaction (PPI) inhibition, a paradigm distinct from classical pharmacological mechanisms of ligand-dependent nuclear receptor modulation. NMR spectroscopy, PPI, and cellular transcription assays show that Nurr1-RXRα transcriptional activation by RXRα ligands is not correlated with classical RXRα agonism but instead correlated with weakening Nurr1-RXRα LBD heterodimer affinity and heterodimer dissociation. Our data inform a model by which pharmacologically distinct RXRα ligands (RXRα homodimer agonists and Nurr1-RXRα heterodimer selective agonists that function as RXRα homodimer antagonists) operate as allosteric PPI inhibitors that release a transcriptionally active Nurr1 monomer from a repressive Nurr1-RXRα heterodimeric complex. These findings provide a molecular blueprint for ligand activation of Nurr1 transcription via small molecule targeting of Nurr1-RXRα.

## Editor's evaluation

This is a fundamental study of the activation process of Nurr1, an orphan nuclear receptor that may be a significant target for the treatment of neurodegenerative disorders. Nurr1 functions as a monomer, but may also heterodimerize with RXRα which represses Nurr1 transcriptional activation. The authors provide compelling evidence for Nurr1 activation through ligand-induced dissociation of an inactive Nurr1-RXRα heterodimer. These data will be important for biochemists and cell biologists working on regulatory / activation mechanisms of nuclear hormone receptors.

## Introduction

Nurr1 (nuclear receptor related 1 protein; NR4A2) is a nuclear receptor (NR) transcription factor that is essential for the development, regulation, and maintenance of several important aspects of mammalian brain development and homeostasis. Nurr1 is critical in the development of dopaminergic neurons that are critical for control of movement that degenerates in Parkinson's disease (PD) (*Decressac et al., 2013*; *Jiang et al., 2005*; *Zetterström et al., 1997*). Recent studies show that Nurr1 is an important factor in the regulation of neuroinflammation and accumulation of amyloid beta that occurs in the pathogenesis of Alzheimer's disease (AD) (*Jeon et al., 2020*; *Moon et al., 2019*). These and other studies implicate small molecule activation of Nurr1 as a potential therapeutic strategy in

aging-associated neurodegenerative and dementia disorders characterized by a loss of neuron function (**Moutinho et al., 2019**).

Although NRs are considered to be ligand-dependent transcription factors, targeting Nurr1 activity with small molecule compounds has remained challenging. Nurr1 contains the conserved NR domain architecture (**Weikum et al., 2018**) including an N-terminal activation function-1 (AF-1) domain, a central DNA-binding domain (DBD), and a C-terminal ligand-binding domain (LBD). The classical model of ligand-induced NR transcriptional activation occurs via ligand binding to an orthosteric pocket located in the internal core of the LBD, which stabilizes the activation function-2 (AF-2 surface) in an active conformation resulting in high affinity coactivation interaction to the NR and recruitment of coactivator complexes resulting in chromatin remodeling and increased gene expression. However, crystal structures have revealed properties of Nurr1 LBD that suggest it may function in an atypical ligand-independent manner. The structures show no classical orthosteric ligand-binding pocket volume as well as a reversed 'charge clamp' AF-2 surface that prevents the LBD from interacting with typical NR transcriptional coregulator proteins (**Wang et al., 2003**). However, solution-state structural studies indicate the Nurr1 LBD is dynamic and can likely expand to bind endogenous ligands (**de Vera et al., 2019**). Although endogenous and synthetic ligands have been reported to bind and/or regulate Nurr1 activity (**Bruning et al., 2019**; **de Vera et al., 2016**; **Kim et al., 2015**; **Rajan et al., 2020**; **Willems and Merk, 2022**), some reported Nurr1 ligands function independent of binding to its LBD (**Munoz-Tello et al., 2020**). These and other observations have directed efforts to discover small molecule modulators of Nurr1 activity through other mechanisms.

An alternative way to modulate Nurr1 transcription activity is through ligands that target its NR heterodimer-binding partner, retinoid X receptor alpha (RXRα; NR2B1). Heterodimerizaton with RXRα represses Nurr1 transcription on monomeric DNA response elements called NBREs (**Aarnisalo et al., 2002**; **Forman et al., 1995**) that are present in the promoter regions of genes that regulate dopaminergic signaling including tyrosine hydroxylase (**Iwawaki et al., 2000**; **Kim et al., 2003**). The observation that classical pharmacological RXRα agonists enhance Nurr1-RXRα transcription on NBREs (**Aarnisalo et al., 2002**; **Forman et al., 1995**; **Perlmann and Jansson, 1995**; **Wallen-Mackenzie et al., 2003**) inspired the discovery of RXRα-binding ligands that display biased activation of Nurr1-RXRα heterodimers over other NR-RXRα heterodimers or RXRα homodimers (**Giner et al., 2015**; **Morita et al., 2005**; **Scheepstra et al., 2017**; **Spathis et al., 2017**; **Sundén et al., 2016**). However, it remains poorly understood how classical RXRα ligands (pharmacological agonists and antagonists) and Nurr1-RXRα selective agonists impact the structure and function of Nurr1-RXRα.

Here, we tested a set of RXRα ligands in structure-function assays to determine their mechanism of action in regulating Nurr1-RXRα transcriptional activity. Unexpectedly, our studies show that ligand activation of Nurr1-RXRα transcription is not associated with classical pharmacological RXRα agonism, which is defined by a ligand-induced increase in coactivator binding to the RXRα LBD resulting in transcriptional activation. Instead, our data support a model whereby Nurr1-RXRα activating ligands weaken Nurr1-RXRα LBD heterodimer affinity via an allosteric protein-protein interaction (PPI) inhibition mechanism and release a transcriptionally active Nurr1 monomer from the repressive Nurr1-RXRα heterodimer.

## Results
### RXRα LBD is sufficient for repression of Nurr1 transcription

To confirm the published observation that RXRα represses Nurr1 transcription (**Aarnisalo et al., 2002**; **Forman et al., 1995**), we performed a transcriptional reporter assay where SK-N-BE(2) neuronal cells were transfected with a full-length Nurr1 expression plasmid, with or without full-length or domain-truncation RXRα expression plasmids, along with a 3xNBRE-luciferase plasmid containing three copies of the monomeric NBRE DNA-binding response element sequence upstream of luciferase gene (**Figure 1a**). Cotransfected RXRα expression plasmid repressed Nurr1 transcription, and RXRα truncation constructs show that the RXRα LBD is both necessary and sufficient for repression of Nurr1 transcription (**Figure 1b**). A slight decrease in Nurr1 transcription is observed with the ΔLBD construct though the effect is not statistically significant. These findings implicate an LBD-driven PPI mechanism with a negligible or minor role for the RXRα DBD, which is consistent with published studies that showed full-length RXRα does not bind to monomeric NBRE sequences via the RXRα DBD, but

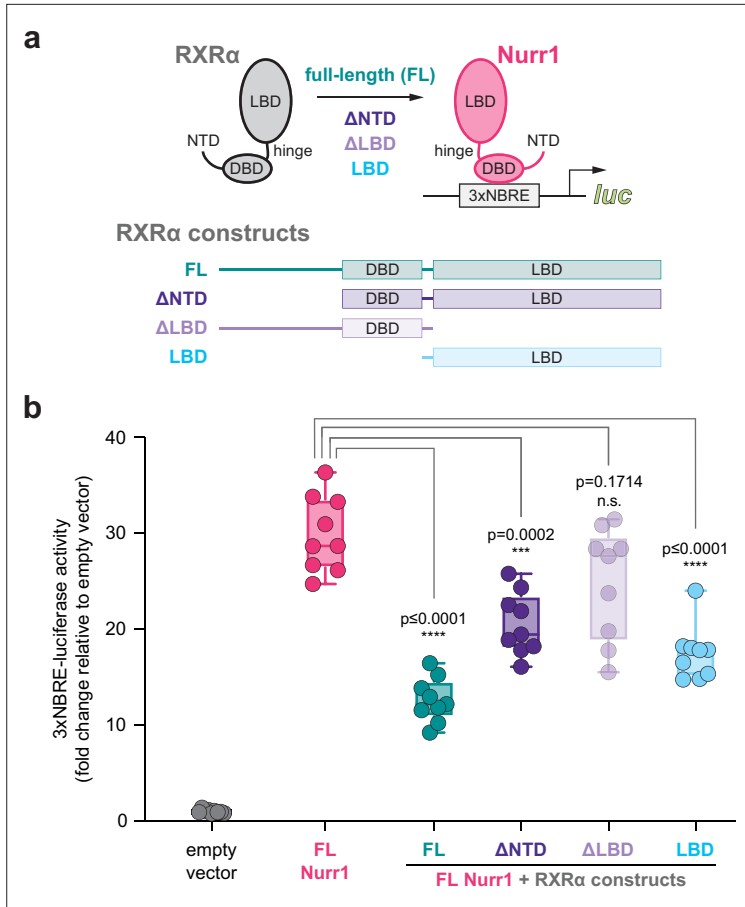

**Figure 1.** Contribution of retinoid X receptor alpha (RXRα) domains on repressing Nurr1 transcription. (**a**) General scheme of the cellular transcriptional reporter assay. (**b**) 3xNBRE-luciferase assay performed in SK-N-BE(2)-C cells; see *Figure 1—source data 1* for data plotted. Data are normalized to empty vector control (n=9 replicates), shown as a box and whiskers plot with boundaries of the box representing the 25th percentile and the 75th percentile, and representative of two or more independent experiments. Statistical testing was performed and p-values were calculated using the Brown-Forsythe and Welch multiple comparisons test of the FL Nurr1 + RXRα constructs conditions relative to FL Nurr1 control condition.

The online version of this article includes the following source data for figure 1:

**Source data 1.** Nurr1+ retinoid X receptor alpha (RXRα) truncated construct luciferase reporter data.

does interact with Nurr1 that is bound to NBRE DNA sequences via a PPI (*Sacchetti et al., 2002*). Furthermore, RXR ligands can activate transcription of a Gal4 DBD-Nurr1 LBD fusion protein, but not a Gal4 DBD-Nurr1 LBD mutant (Nurr1[dim]) that cannot heterodimerize with RXRα (*Aarnisalo et al., 2002*; *Wallen-Mackenzie et al., 2003*), which further implicates the RXRα LBD in repression of Nurr1 LBD-mediated transcription.

## RXRα ligands display graded Nurr1-RXRα transcriptional activation

We assembled a set of 14 commercially available RXRα-binding ligands (*Figure 2*) described in the literature as pharmacological RXRα agonists and antagonists; a mixed activity PPARγ-RXR selective modulator (LG100754) that antagonizes RXR homodimers but agonizes PPAR-RXR and RAR-RXR heterodimers (*Lala et al., 1996*) and two RXRα-binding compounds (BRF110 and HX600) described as selective agonists of Nurr1-RXRα heterodimers (*Morita et al., 2005*; *Spathis et al., 2017*). To determine how the ligands influence Nurr1-RXRα transcription, we performed a transcriptional reporter assay where SK-N-BE(2) neuronal cells were transfected with full-length Nurr1 and RXRα expression plasmids and the 3xNBRE-luc plasmid then treated with compound or DMSO control (*Figure 3a*). Compounds reported as RXRα agonists increase Nurr1-RXRα transcription (*Figure 3b*), and among

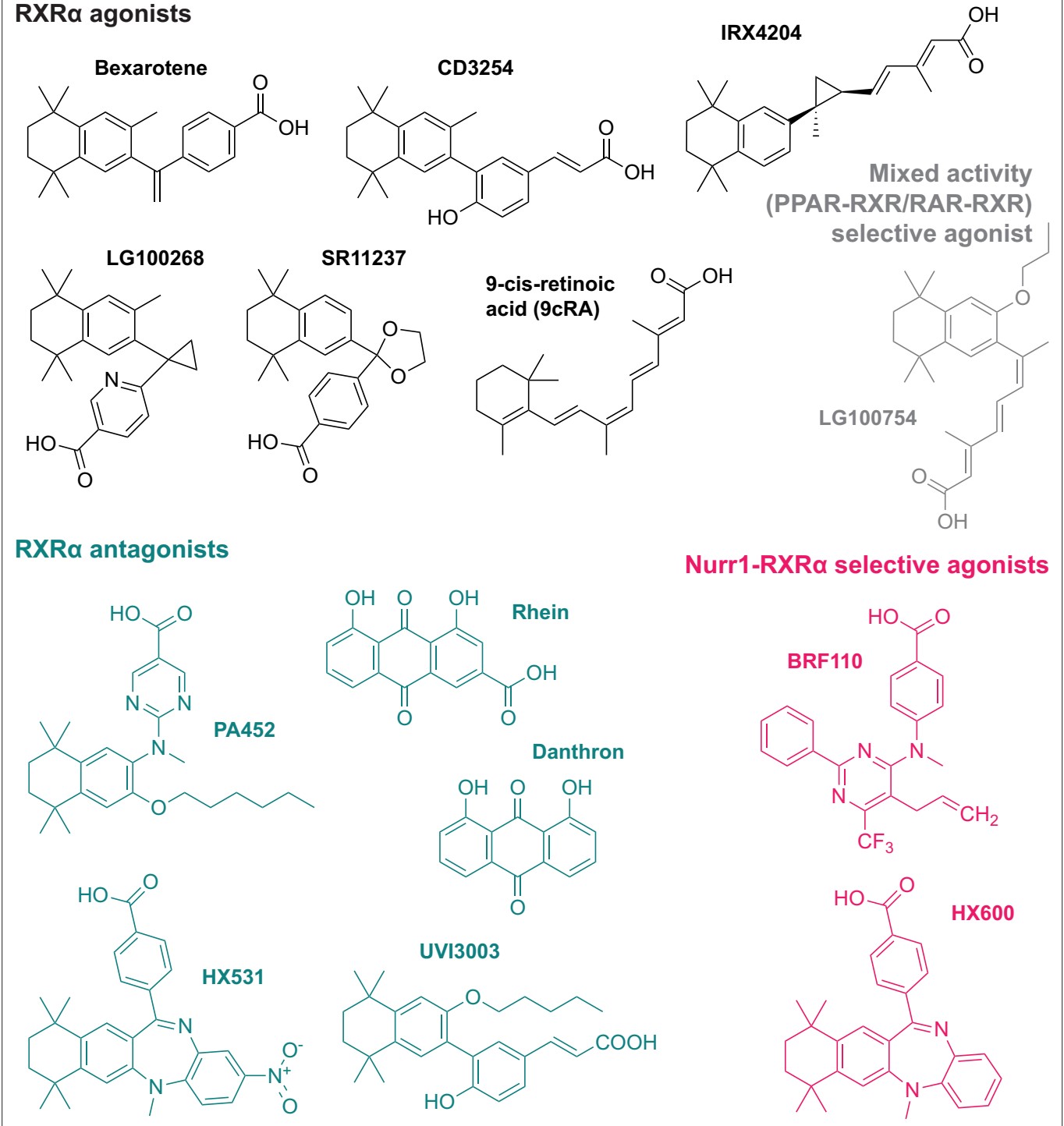

**Figure 2.** Retinoid X receptor alpha (RXRα) ligands used in this study. Grouped by pharmacological phenotype, the set includes ligands that classically activate (agonists) or block (antagonists) activation of RXRα homodimers; a mixed activity modulator (LG100754) that antagonizes RXRα homodimers and activates PPARγ-RXRα and RAR-RXRα heterodimers; and two selective activators of Nurr1-RXRα heterodimers (BRF110 and HX600).

the most efficacious agonist ligands includes endogenous metabolite 9-*cis*-retinoic acid (9cRA) and bexarotene, the latter of which was reported to display biased activation of Nurr1-RXR heterodimers over RXR homodimers (*McFarland et al., 2013*). RXRα antagonists showed relatively no change or slightly decreased transcription of Nurr1-RXRα. The two Nurr1-RXRα selective agonists, BRF110 and HX600, displayed the highest activity of all compounds tested. Taken together, the transcriptional

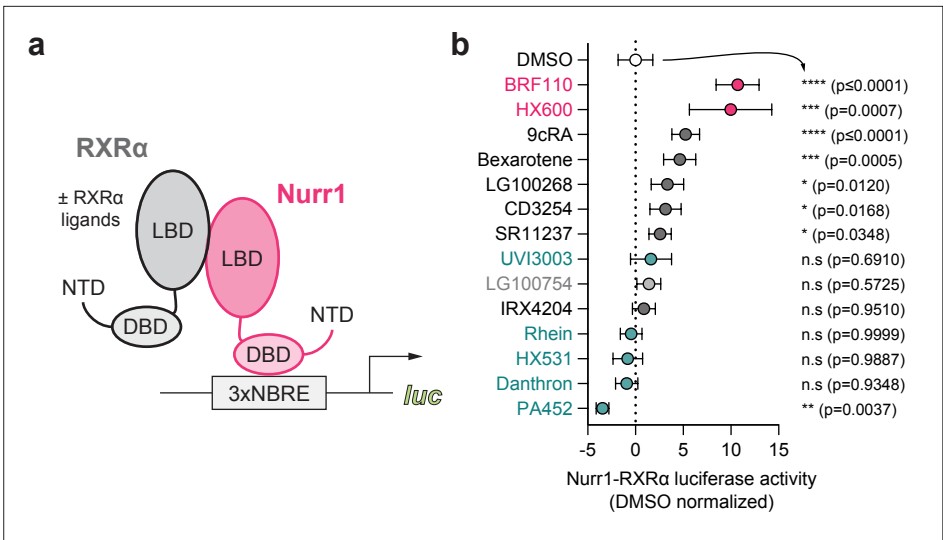

**Figure 3.** Effect of retinoid X receptor alpha (RXRα)-binding ligands on Nurr1-RXRα transcription. (**a**) General scheme of the Nurr1-RXRα/3xNBRE-luciferase cellular transcriptional reporter assay. (**b**) Nurr1-RXRα/3xBNRE-luciferase transcriptional reporter assay performed in SK-N-BE(2)-C cells treated with RXRα ligand (1 μM) or DMSO (dotted line); see *Figure 3—source data 1* for data plotted. Data are normalized to DMSO (n=9 replicates), represent the mean ± s.d., and representative of two or more independent experiments. Statistical testing was performed and p-values were calculated using the Brown-Forsythe and Welch multiple comparisons test relative to DMSO control treated condition.

The online version of this article includes the following source data for figure 3:

**Source data 1.** Retinoid X receptor alpha (RXRα) ligand treated Nurr1-RXRα/3xNBRE3-luciferase reporter data.

reporter data obtained indicates this RXRα ligand set influences Nurr1-RXRα transcription via a graded activation mechanism.

## Nurr1-RXRα activation is not correlated with pharmacological RXRα agonism

Agonist binding to the NR LBD stabilizes an active conformation that facilitates coactivator protein interaction at the AF-2 surface via a 'charge clamp' mechanism (*Savkur and Burris, 2004*) that is important for binding LXXLL-containing motifs present within coactivator proteins resulting in an increase in transcription (*Kojetin and Burris, 2013*). The Nurr1 LBD does not interact with canonical coregulator proteins because it contains a reversed 'charge clamp' in its AF-2 surface (*Wang et al., 2003*), implicating ligand-dependent coactivator interaction with RXRα in the mechanism of Nurr1-RXRα activation.

To determine if there is a correlation between Nurr1-RXRα transcriptional agonism and increased coactivator recruitment to RXRα that results in transcriptional activation within the RXRα ligand set, we performed a time-resolved fluorescence resonance energy transfer (TR-FRET) biochemical assay (*Figure 4a*) to assess how the compounds affect interaction between the RXRα LBD and a coregulator peptide derived from a cognate coactivator protein PGC-1α (*Delerive et al., 2002*; *Figure 4b*). We also performed a transcriptional reporter assay to determine if there is a correlation between RXRα homodimer-mediated transcription (*Figure 4c*) and Nurr1-RXRα transcription for the RXRα ligand set. We cotransfected HEK293T cells with a full-length RXRα expression plasmid along with a plasmid containing three copies of the dimeric RXR DNA-binding response element sequence upstream of luciferase gene (3xDR1-luc) then treated the cells with RXRα ligand or DMSO control (*Figure 4d*). In this analysis, we did not include two RXRα ligands, danthron and rhein, which are colored and cause interference in TR-FRET experiments.

We previously showed that ligands displaying graded pharmacological PPARγ agonism show a strong correlation between coactivator peptide recruitment to the LBD in a TR-FRET biochemical assay and cellular transcription of full-length PPARγ (*Shang et al., 2019*). However, no significant

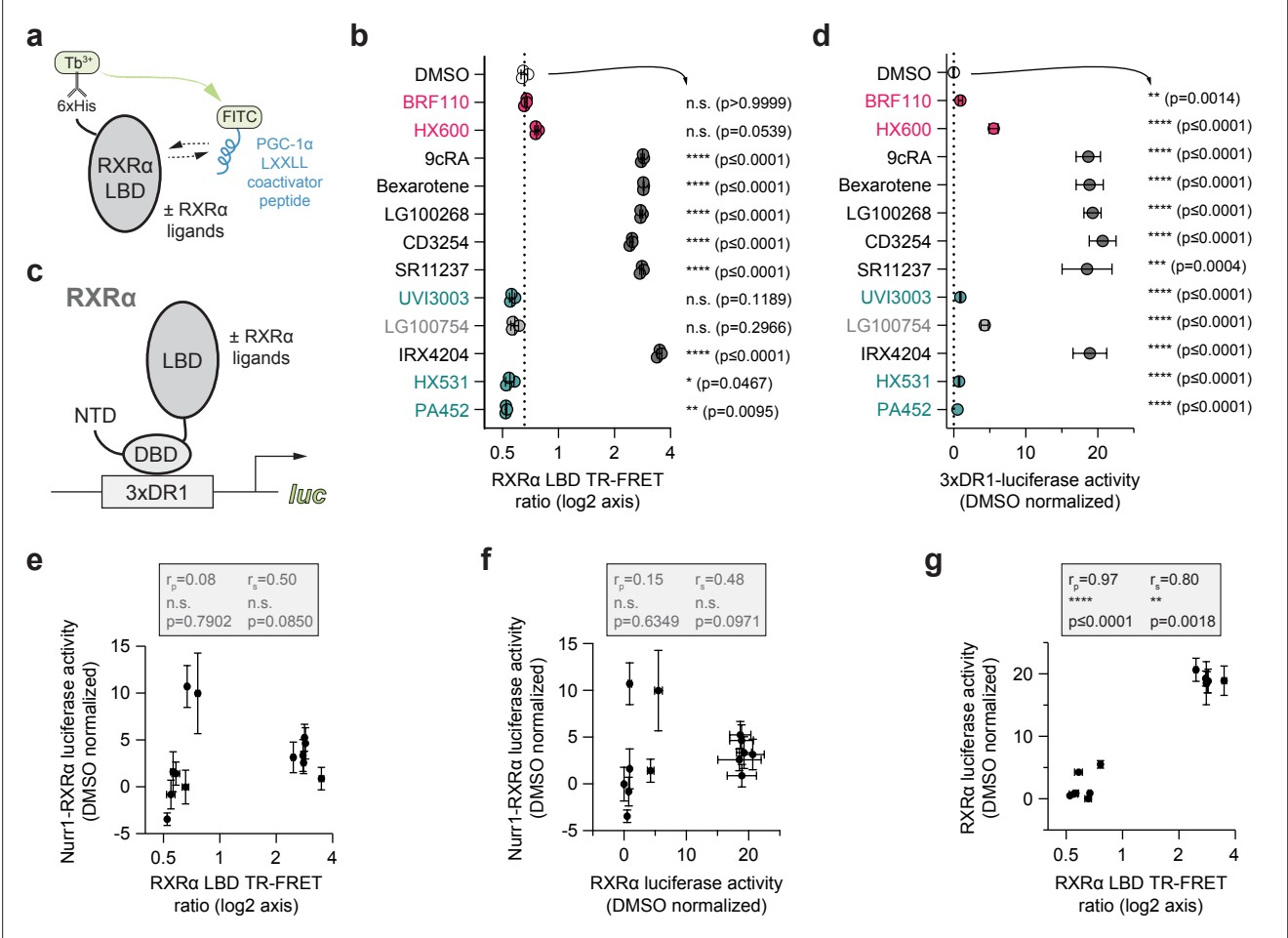

**Figure 4.** Compound profiling for pharmacological retinoid X receptor alpha (RXRα) agonism and correlation to Nurr1-RXRα agonism. (**a**) General scheme of the RXRα ligand-binding domain (LBD) time-resolved fluorescence resonance energy transfer (TR-FRET) coactivator peptide interaction assay. (**b**) TR-FRET ratio measured in the presence of DMSO (dotted line) or compound (2–4 μM); see *Figure 4—source data 1* for data plotted. Data are normalized to DMSO control (n=3 biological replicates), represent the mean ± s.d., representative of two or more independent experiments. Statistical testing was performed and p-values were calculated using ordinary one-way ANOVA tests for multiple comparisons with Dunnett corrections relative to DMSO control treated condition. (**c**) General scheme of the RXRα/3xDR1-luciferase cellular transcriptional reporter assay. (**d**) RXRα/3xDR1-luciferase transcriptional reporter assay performed in HEK293T cells treated with compound (1 μM) or DMSO control (dotted line); see *Figure 4—source data 2* for data plotted. Data normalized to DMSO (n=6 replicates), represent the mean ± s.d., and representative of two or more independent experiments. Statistical testing was performed and p-values were calculated using the Brown-Forsythe and Welch multiple comparisons test relative to DMSO control treated condition. (**e**) Correlation plot of RXRα LBD TR-FRET data vs. Nurr1-RXRα cellular transcription data. (**f**) Correlation plot of RXRα transcriptional reporter data vs. Nurr1-RXRα cellular transcription data. (**g**) Correlation plot of RXRα transcriptional reporter data vs. RXRα LBD TR-FRET data. Pearson ($r_p$) and Spearman ($r_s$) correlation coefficients and statistical significance testing are reported above the correlation plots.

The online version of this article includes the following source data for figure 4:

**Source data 1.** Retinoid X receptor alpha (RXRα) ligand treated RXRα ligand-binding domain (LBD) time-resolved fluorescence resonance energy transfer (TR-FRET) coactivator interaction data.

**Source data 2.** Retinoid X receptor alpha (RXRα) ligand treated RXRα/3xDR1-luciferase reporter data.

correlation was observed between coactivator peptide recruitment to the RXRα LBD and Nurr1-RXRα transcription for the RXRα ligand set (*Figure 4e*). Furthermore, no significant correlation is observed between RXRα homodimer transcription and Nurr1-RXRα transcription (*Figure 4f*). In contrast, a strong correlation is observed between RXRα LBD TR-FRET data and full-length RXRα transcription (*Figure 4g*) where ligands that increase PGC-1α coactivator peptide interaction to the RXRα LBD activate transcription of full-length RXRα in cells, similar to what we observed for PPARγ agonists with graded agonist transcriptional activity (*Shang et al., 2019*).

The two selective Nurr1-RXRα activating compounds, BRF110 and HX600, function as a pharmacological antagonist and weak/partial graded agonist in the RXRα homodimer TR-FRET and transcriptional reporter assay, respectively. These findings suggest that the mechanism by which the RXRα ligand set influences Nurr1-RXRα transcription may occur independent of coregulator recruitment to the RXRα LBD and RXRα-mediated transcription—indicating that ligand-dependent RXRα homodimer modulation and Nurr1-RXRα heterodimer modulation may function through distinct mechanisms.

## Nurr1-RXRα activation is correlated with weakening LBD heterodimer affinity

Although ligand binding to NRs typically influences coregulator interaction to the AF-2 surface in the LBD, studies have reported that ligand binding can also weaken or strengthen NR LBD

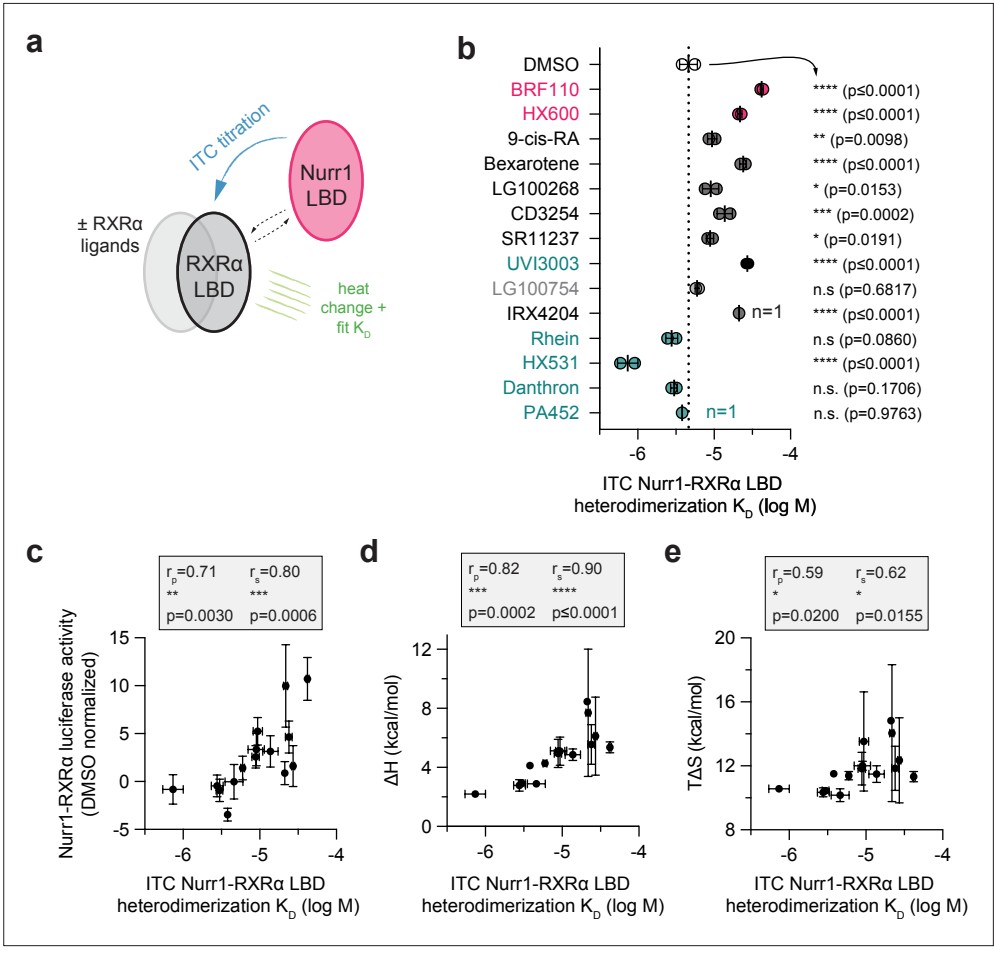

**Figure 5.** Compound profiling for effects on Nurr1-retinoid X receptor alpha (RXRα) ligand-binding domain (LBD) heterodimer affinity and correlation to Nurr1-RXRα agonism. (**a**) General scheme of the Nurr1-RXRα LBD isothermal titration calorimetry (ITC) experiment. (**b**) Nurr1-RXRα LBD heterodimer affinities (log M) in the presence of DMSO (dotted line) or compound determined from the fit of the ITC data (n=2 replicates, except n=1 for IRX4204 and PA452) and represent the mean ± s.d.; see *Table 1* and *Figure 5—source data 1* for data plotted. Statistical testing was performed and p-values were calculated using ordinary one-way ANOVA relative to DMSO control treated condition. (**c**) Correlation plot of ITC determined Nurr1-RXRα LBD heterodimer $K_D$ log M values vs. Nurr1-RXRα cellular transcription data. (**d**) Correlation plot of ITC determined Nurr1-RXRα LBD heterodimer $K_D$ log M values vs. fitted binding enthalpy (ΔH, which is the ΔHAB component of the homodimer competition model; see Materials and methods section for details). (**e**) Correlation plot of ITC determined Nurr1-RXRα LBD heterodimer $K_D$ log M values vs. calculated binding entropy (TΔS). Pearson ($r_p$) and Spearman ($r_s$) correlation coefficients and statistical significance testing are reported above the correlation plot.

The online version of this article includes the following source data for figure 5:

**Source data 1.** Raw isothermal titration calorimetry (ITC) thermograms and fitted data.

homodimerization and heterodimerization and confer selectivity (*Kilu et al., 2021*; *Powell and Xu, 2008*; *Rehó et al., 2020*; *Tamrazi et al., 2002*). To determine how the RXRα ligand set influences Nurr1-RXRα LBD heterodimerization affinity, we performed isothermal titration calorimetry (ITC) studies where we titrated Nurr1 LBD into apo/ligand-free or ligand-bound RXRα LBD (*Figure 5a*) and fitted the data to a homodimer competition model that incorporates apo-RXRα LBD homodimerization affinity (see Materials and methods) and dissociation of RXRα LBD homodimers to obtain Nurr1-RXRα LBD heterodimerization affinity (*Table 1* and *Figure 5b*). One limitation of this ITC data analysis is the assumption that the RXRα LBD homodimer affinity is the same for apo RXRα LBD and the various ligand-bound states. Determining ligand-bound RXRα LBD homodimer $K_D$ values using ITC via the dimer dissociation dilution (*McPhail and Cooper, 1997*) would likely be confounded by ligand dissociation. It is also possible that RXRα ligands could change RXRα LBD homodimer affinity, or potentially change the RXRα LBD dimer equilibrium toward a tetrameric form where the Nurr1 LBD ITC titration could include a component that dissociates a ligand-bound RXRα LBD homotetramer (*Chen et al., 1998*; *Gampe et al., 2000*; *Zhang et al., 2011a*). Despite these limitations, a significant correlation between Nurr1-RXRα LBD heterodimerization affinity and Nurr1-RXRα transcription where RXRα ligands that weaken heterodimerization affinity show higher Nurr1-RXRα transcription (*Figure 5c*).

To gain additional insight into the thermodynamic mechanism of ligand-induced Nurr1-RXRα LBD heterodimer dissociation, we analyzed the thermodynamic parameters from the fitted ITC data (*Table 1*). The change in binding enthalpy (ΔH) upon heterodimerization generally shows an endothermic (positive) profile for all conditions, whereas the change in entropy (TΔS) component shows a favorable (positive) profile for all conditions. Correlation analysis reveals a more significant correlation between the Nurr1-RXRα LBD heterodimerization binding affinity and the ΔH binding component (*Figure 5d*) compared to the ΔS binding component (*Figure 5e*). Moreover, compounds that decrease Nurr1-RXRα LBD heterodimer binding affinity show an increasing (less favorable) ΔH component and an increasing (more favorable) ΔS component.

## Nurr1-RXRα activation is correlated with dissociation of Nurr1 LBD monomer

To obtain structural insight into the consequence of ligand-induced weakening of Nurr1-RXRα LBD heterodimer and the relationship to transcriptional activation of Nurr1-RXRα, we performed protein NMR structural footprinting analysis. We collected 2D [$^1$H,$^{15}$N]-TROSY-HSQC NMR data of $^{15}$N-labeled Nurr1 LBD in the monomer form or heterodimerized with RXRα LBD. Among the well-resolved Nurr1 LBD peaks in the 2D NMR data that show clear and discernible shifts going from monomeric Nurr1 LBD to the Nurr1-RXRα LBD heterodimer (*Figure 6—figure supplement 1*) include the NMR peak for Thr411 (*Figure 6a*). Addition of compounds that activate Nurr1-RXRα transcription and weaken Nurr1-RXRα LBD heterodimer affinity result in the appearance of two Thr411 NMR peaks with chemical shift values corresponding to the heterodimer and Nurr1 monomer populations. In contrast, compounds that display little to no effect on Nurr1-RXRα transcription and do not significantly alter Nurr1-RXRα LBD heterodimer affinity show either a single NMR peak corresponding to the heterodimer population or a lower Nurr1 monomer population on average. These data indicate that Nurr1-RXRα activating ligands perturb the Nurr1-RXRα LBD heterodimer conformational ensemble toward a monomeric Nurr1 LBD population in slow exchange on the NMR time scale.

To gain additional insight into how ligand-dependent changes in Nurr1-RXRα LBD heterodimerization affinity influences the NMR data, we performed NMR lineshape simulations. Simulated 1D spectra of the $^1$H$_N$ dimension of the Thr411 NMR peaks were calculated using a binding model that accounts for an RXRα LBD homodimer component that dissociates into monomers that are capable of binding Nurr1 LBD (see Materials and methods for details). The simulations show that weakening Nurr1-RXRα LBD heterodimerization affinity, for example upon binding an RXRα ligand, shifts the equilibrium from the Nurr1-RXRα LBD heterodimeric (*hd*) population to Nurr1 LBD monomer (*m*) population (*Figure 6b*), consistent with our experimentally collected 2D NMR data.

We calculated the relative population of monomeric Nurr1 LBD dissociated from the Nurr1-RXRα LBD heterodimer by measuring the peak intensity of the monomeric (*m*) and heterodimer (*hd*) 2D NMR peaks for Thr411 from two replicate measurements using different batches of protein (*Figure 6c*). One limitation of this NMR analysis is that the monomeric and heterodimeric NMR peak intensities may over- or underestimate the relative population sizes given that NMR peak lineshapes

**Table 1.** Fitted and calculated isothermal titration calorimetry (ITC) Nurr1-RXRα binding affinity and thermodynamic parameters.

| Ligand | log KD (M) | | | | ΔH (kcal/mol) | | | | TΔS (kcal/mol) | | | |
|---|---|---|---|---|---|---|---|---|---|---|---|---|
| | Average | s.d. | n | Replicates | Average | s.d. | n | Replicates | Average | s.d. | n | Replicates |
| DMSO | −5.335 | 0.118 | 2 | −5.418 / −5.252 | 2.900 | 0.240 | 2 | 3.070 / 2.730 | 10.179 | 0.401 | 2 | 10.462 / 9.895 |
| BRF110 | −4.376 | 0.020 | 2 | −4.390 / −4.362 | 5.365 | 0.361 | 2 | 5.110 / 5.620 | 11.335 | 0.334 | 2 | 11.099 / 11.571 |
| HX600 | −4.662 | 0.028 | 2 | −4.681 / −4.642 | 7.695 | 4.320 | 2 | 4.640 / 10.750 | 14.054 | 4.283 | 2 | 11.026 / 17.083 |
| 9cRA | −5.026 | 0.060 | 2 | −4.984 / −5.068 | 5.110 | 0.962 | 2 | 5.790 / 4.430 | 13.539 | 3.105 | 2 | 12.589 / 11.344 |
| Bexarotene | −4.619 | 0.044 | 2 | −4.588 / −4.650 | 5.555 | 1.336 | 2 | 4.610 / 6.500 | 11.857 | 1.396 | 2 | 10.870 / 12.844 |
| LG100268 | −5.044 | 0.106 | 2 | −5.119 / −4.969 | 5.135 | 0.148 | 2 | 5.240 / 5.030 | 12.017 | 0.294 | 2 | 12.224 / 11.809 |
| CD3254 | −4.860 | 0.099 | 2 | −4.930 / −4.790 | 4.865 | 0.389 | 2 | 5.140 / 4.590 | 11.495 | 0.524 | 2 | 11.866 / 11.125 |
| SR11237 | −5.053 | 0.047 | 2 | −5.087 / −5.020 | 4.940 | 0.962 | 2 | 5.620 / 4.260 | 11.834 | 1.026 | 2 | 12.560 / 11.109 |
| UVI3003 | −4.565 | 0.021 | 2 | −4.580 / −4.550 | 6.125 | 2.638 | 2 | 7.990 / 4.260 | 12.353 | 2.666 | 2 | 14.238 / 10.467 |
| LG100754 | −5.222 | 0.026 | 2 | −5.204 / −5.240 | 4.270 | 0.226 | 2 | 4.110 / 4.430 | 11.394 | 0.261 | 2 | 11.209 / 11.579 |
| IRX4204 | −4.673 | 0.000 | 1 | −4.673 | 8.460 | 0.000 | 1 | 8.460 | 14.835 | 0.000 | 1 | 14.835 |
| Rhein | −5.554 | 0.077 | 2 | −5.499 / −5.609 | 2.780 | 0.396 | 2 | 3.060 / 2.500 | 10.357 | 0.290 | 2 | 10.563 / 10.152 |
| HX531 | −6.132 | 0.133 | 2 | −6.226 / −6.038 | 2.205 | 0.021 | 2 | 2.190 / 2.220 | 10.571 | 0.160 | 2 | 10.684 / 10.457 |
| Danthron | −5.524 | 0.043 | 2 | −5.494 / −5.554 | 2.920 | 0.170 | 2 | 3.040 / 2.800 | 10.456 | 0.111 | 2 | 10.535 / 10.378 |
| PA452 | −5.417 | 0.000 | 1 | −5.417 | 4.120 | 0.000 | 1 | 4.120 | 11.510 | 0.000 | 1 | 11.510 |

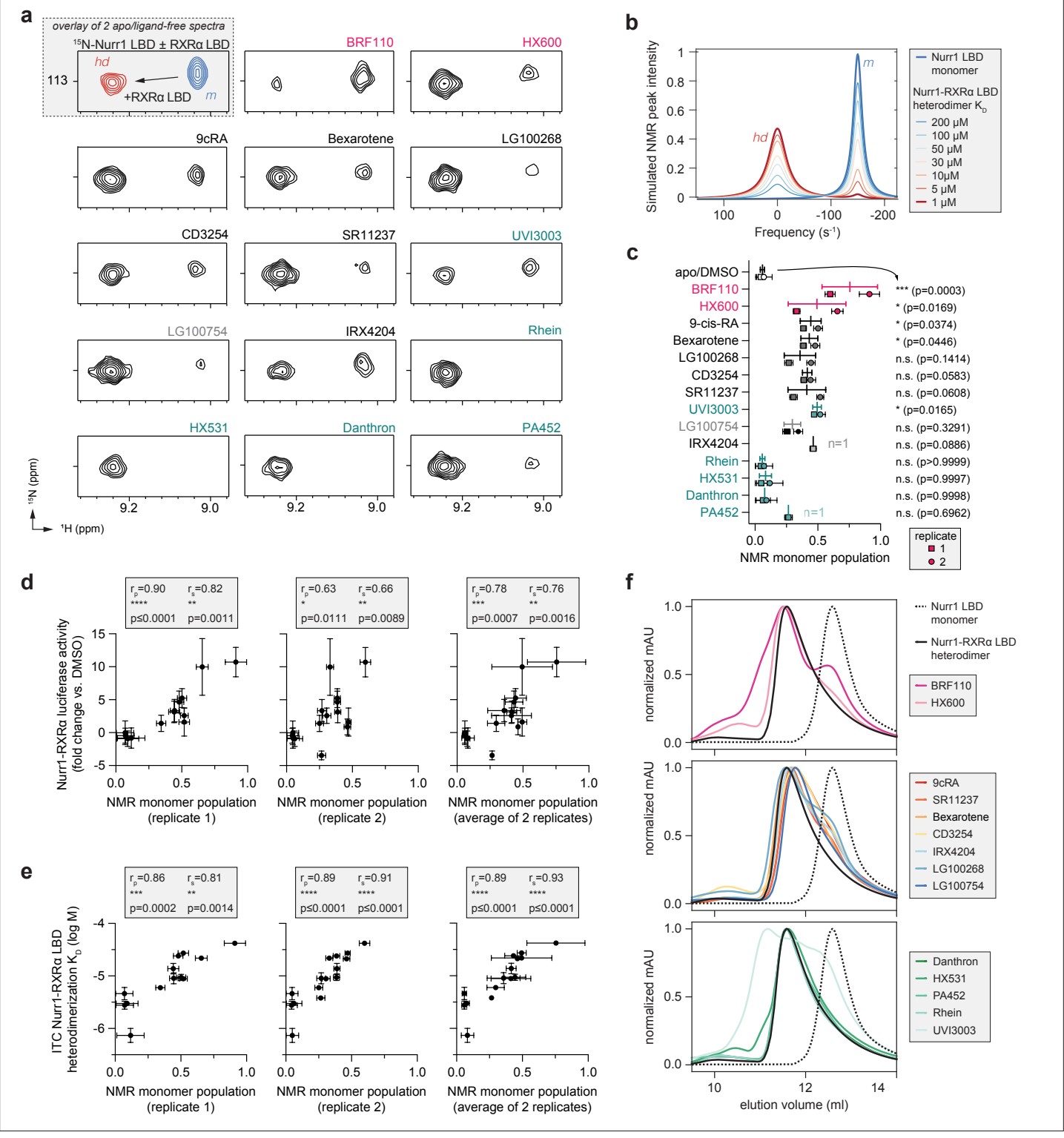

**Figure 6.** Compound profiling for effects on Nurr1-retinoid X receptor alpha (RXRα) ligand-binding domain (LBD) conformational properties in solution and correlation to Nurr1-RXRα agonism. (**a**) 2D [$^1$H,$^{15}$N]-TROSY HSQC data of $^{15}$N-labeled Nurr1 LBD heterodimerized with unlabeled RXRα LBD in the presence of RXRα ligands focused on the NMR peak of Thr411. The upper inset shows an overlay of $^{15}$N-labeled Nurr1 LBD monomer (200 μM) vs. $^{15}$N-labeled Nurr1 LBD-unlabeled RXRα LBD heterodimer (1:2 molar ratio) to demonstrate the shift of the Thr411 peak between monomer (**m**) and heterodimer (**hd**) forms; see *Figure 6—figure supplement 1* for full spectral overlays. (**b**) Simulated $^1$H NMR lineshape analysis of Nurr1 LBD residue Thr411 showing the influence of ligand-induced weakening of Nurr1-RXRα LBD heterodimerization affinity; see *Figure 6—source code 1* for

*Figure 6 continued on next page*

*Figure 6 continued*

calculation input files. (**c**) NMR estimated Nurr1 LBD monomer populations from the 2D NMR data (n=2 replicates, except n=1 for IRX4204 and PA452) and lines above the replicate values represent the mean ± s.d.; see *Figure 6—source data 1* for data plotted. Statistical testing was performed and p-values were calculated using ordinary one-way ANOVA relative to apo/DMSO control treated condition. (**d**) Correlation plot of Nurr1-RXRα cellular transcription data vs. NMR estimated Nurr1 LBD monomer populations. (**e**) Correlation plot of ITC determined Nurr1-RXRα LBD heterodimer $K_D$ log M values vs. NMR estimated Nurr1 LBD monomer populations. Pearson ($r_p$) and Spearman ($r_s$) correlation coefficients and statistical significance testing are reported above the correlation plots. (**f**) Analytical size exclusion chromatography (SEC) analysis of Nurr1-RXRα in the presence of RXRα ligands (solid colored lines) relative to Nurr1 LBD monomer (dotted black line) and Nurr1-RXRα LBD heterodimer (solid black line); see *Figure 6—figure supplement 2* for all SEC data organized by ligand.

The online version of this article includes the following source data, source code, and figure supplement(s) for figure 6:

**Source code 1.** Input files for NMR LineShapeKin simulated NMR data analysis in MATLAB (two input files and one readme file).

**Source data 1.** Retinoid X receptor alpha (RXRα) ligand treated Nurr1-RXRα ligand-binding domain (LBD) NMR-observed monomer species.

**Figure supplement 1.** Full overlay of 2D [$^1$H,$^{15}$N]-TROSY-HSQC data of $^{15}$N-labeled Nurr1 ligand-binding domain (LBD) (200 µM) in monomeric (**m**) and heterodimer (hd) forms with retinoid X receptor alpha (RXRα) LBD (400 µM).

**Figure supplement 2.** Analytical size exclusion chromatography (SEC) profiles of Nurr1 ligand-binding domain (LBD) (monomer), retinoid X receptor alpha (RXRα) LBD (homodimer and homotetramer), and Nurr1-RXRα LBD (heterodimer) with RXRα ligands present in the RXRα LBD-containing conditions (homodimer or Nurr1-RXRα heterodimer).

**Figure supplement 3.** SDS-PAGE analysis of fractions collected from size exclusion chromatography (SEC) analysis of Nurr1-retinoid X receptor alpha (RXRα) ligand-binding domain (LBD) heterodimer in the presence of BRF110.

**Figure supplement 3—source data 1.** Full raw unedited gel, without and with annotation (two JPG files).

of the individual monomeric and heterodimer states (molecular sizes effects), and the rate of chemical exchange between these states, can also affect peak lineshapes as well as binding kinetics and affinity. Despite this limitation, a significant correlation is observed between NMR-observed Nurr1 LBD monomeric populations and Nurr1-RXRα transcription (*Figure 6d*) and ITC-determined Nurr1-RXRα LBD heterodimerization affinity (*Figure 6e*), indicating a role for dissociation of monomeric Nurr1 LBD from Nurr1-RXRα LBD heterodimer in the mechanism of action of the RXRα ligand set. Notably, one of the replicate NMR measurements shows a more significant correlation to Nurr1-RXRα transcription, which may have at least two origins: errors in concentration between two different batches of protein used in the replicate measurements, and two ligands were not included in one of the replicate measurements—excluding these ligands from the analysis of replicate #2 measurements improves the correlation with $r_p = 0.77$ (p=0.0021) and $r_s = 0.75$ (p=0.0042).

To corroborate our ligand-dependent heterodimer dissociation findings, we performed size exclusion chromatography (SEC) experiments to determine how the compounds influence the oligomeric state of the Nurr1-RXRα LBD heterodimer (*Figure 6g* and *Figure 6—figure supplement 2*). The Nurr1-RXRα selective agonist, BRF110, shows the largest effect in dissociating a Nurr1 LBD monomer species in the SEC and NMR studies and is overall the most efficacious compound in activating Nurr1-RXRα transcription. The other Nurr1-RXRα selective compound, HX600, and the classical RXRα agonists dissociate a Nurr1 LBD monomer species although to a lower degree compared to BRF110. Most pharmacological RXRα antagonists we studied do not dissociate a Nurr1 LBD monomer species except for UVI3003, which displays a unique SEC profile that includes a monomeric Nurr1 LBD species and at least two species with a larger hydrodynamic radius (*Figure 6—figure supplement 2*). One higher order species is consistent with the SEC profile of an RXRα LBD homodimer or Nurr1-RXRα LBD heterodimer, and another with a larger hydrodynamic radius that is more compact than RXRα homotetramer, suggesting it may be a UVI3003-bound RXRα LBD dimer population. Notably, our structure-function profiling data indicate UVI3003 functions similar to the Nurr1-RXRα selective ligands (BRF110 and HX600) in that it activates Nurr1-RXRα transcription while antagonizing RXRα homodimers.

The SEC data provide insight into the potential allosteric mechanism(s) by which compounds in the RXRα ligand set influence Nurr1-RXRα heterodimerzation and transcription via effects on RXRα LBD. In the absence of ligand, RXRα exists as a mixture of homodimers and homotetramers (*Kersten et al., 1995a*), which can be stabilized as dimers upon agonist binding (*Kersten et al., 1995b*) or tetramers upon antagonist binding (*Zhang et al., 2011b*). In our SEC data, the most efficacious compounds not only free a Nurr1 LBD monomeric population, but also increase a higher order species with an elution volume consistent with a RXRα LBD homotetramer population. SDS-PAGE analysis confirms this observation and reveals that the higher order species observed when BRF110 is added to Nurr1-RXRα

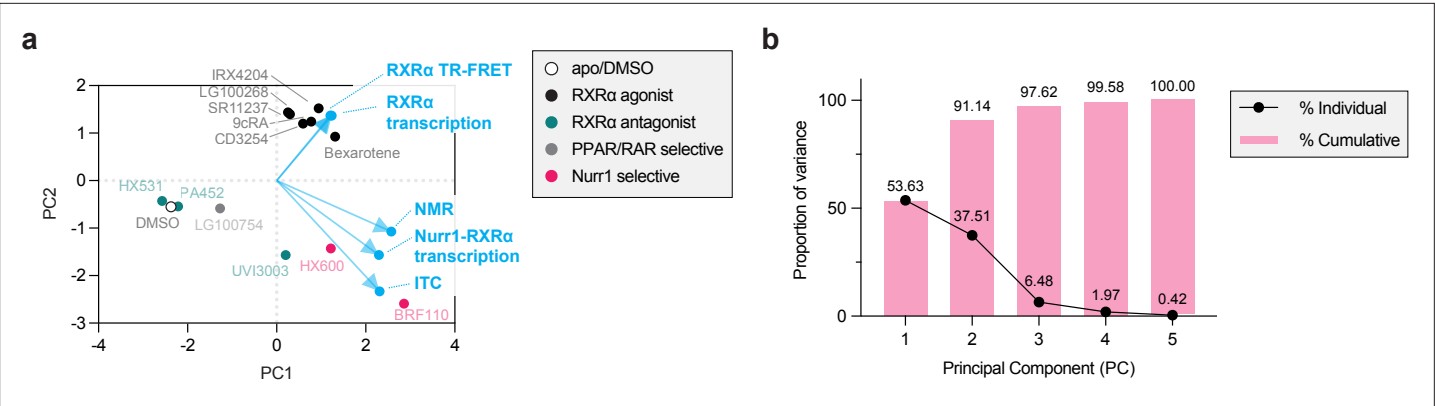

**Figure 7.** Principal component analysis (PCA) of experimental data reveals Nurr1-retinoid X receptor alpha (RXRα) agonism and classical RXRα agonism are uncorrelated. (**a**) 2D biplot containing the loadings and PC scores of the first two PCs. PC scores of the ligands are colored according to pharmacological activity (see legend) whereas the loadings (experimental variables) are colored light blue. (**b**) A proportion of variance plot revealing the amount of variance described by each PC individually and cumulatively.

LBD heterodimers contains RXRα LBD only (*Figure 6—figure supplement 3*). Furthermore, these compounds change the appearance of species in the SEC data with elution properties similar to Nurr1-RXRα heterodimer and RXRα homodimers, suggesting that they may change the homodimer/heterodimer complex conformation and/or species equilibrium. Thus, the allosteric mechanism may involve ligand-induced stabilization of RXRα oligomeric species that inhibit or are less favorable for Nurr1 heterodimerization.

## PCA reveals data features associated with Nurr1-RXRα agonism

Finally, we performed principal component analysis (PCA) using all the experimental data collected as input to determine the data features most associated with transcriptional activation of the 3xNBRE-luciferase reporter by Nurr1-RXRα in an unbiased manner (*Figure 7a*). Two of the ligands, danthron and rhein, were not included in the analysis because we did not have RXRα LBD TR-FRET and RXRα homodimer transcriptional reporter data (vide infra). Approximately 90% of the variance in the data can be explained by the two PCs (*Figure 7b*). Several features are notable in the plot of the PCA scores and loadings, the latter of which represent correlation between the experimental data and PCs. The ligands cluster in groups consistent with their known pharmacological phenotypes. The loadings show that Nurr1-RXRα transcription, ITC-determined Nurr1-RXRα heterodimer affinity, and NMR-determined Nurr1 monomeric populations are clustered together and clustered along with the Nurr1-RXRα selective ligands, meaning these features of the data are positively correlated with ligand-dependent Nurr1-RXRα heterodimer dissociation. In contrast, RXRα LBD TR-FRET coactivator interaction and RXRα LBD homodimer transcription is positively correlated with classical RXRα agonists, whereas RXRα antagonists are found opposite to this loading trajectory. Furthermore, the loading projections show that the features of classical RXRα agonism and selective Nurr1-RXRα agonism are uncorrelated since the loadings trajectories associated with these features, and their associated ligand clusters, form a perpendicular (~90°) angle.

## Discussion

Although Nurr1 is a promising drug target for aging-associated neurodegenerative disorders, uncertainty in the druggability of its LBD has motivated research into alternative approaches to influence Nurr1 transcription. Nurr1-RXRα heterodimer activation via RXRα-binding ligands has emerged as a promising modality to develop neuroprotective therapeutic agents or adjuvants to other therapies. Several RXRα ligands that increase Nurr1-RXRα activation have progressed to clinical trials of PD and AD including bexarotene and IRX4204 (*McFarland et al., 2013*; *Wang et al., 2016*). Understanding how these and other Nurr1-RXRα activating ligands function on the molecular level may provide a blueprint to develop more efficacious and potent compounds.

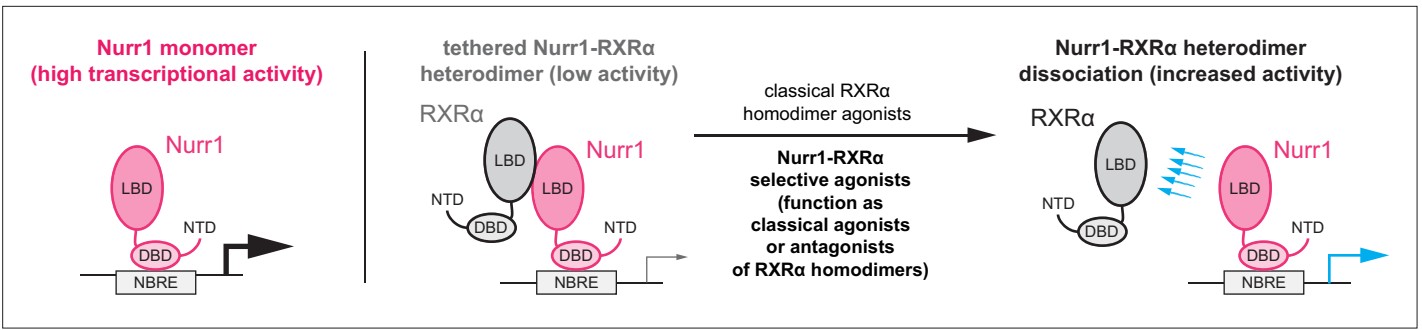

**Figure 8.** Data-informed model for activation of Nurr1-retinoid X receptor alpha (RXRα) transcription by RXRα ligands.

Classical pharmacological modulation of NR transcription by agonists, antagonists, or inverse agonists that activate, block, or repress transcription typically function by enhancing coactivator protein interaction, blocking coregulator interaction, or enhancing corepressor interaction, respectively (*Gronemeyer et al., 2004*). Our studies demonstrate that Nurr1-RXRα activation occurs through an LBD PPI inhibition heterodimer dissociation mechanism that is distinct from the classical pharmacological properties of NRs ligands. This activation mechanism explains why pharmacological RXRα agonists (e.g. bexarotene and IRX4204) and Nurr1-RXRα selective compounds that function as pharmacological RXRα antagonists (e.g. BRF110 and HX600) both activate Nurr1-RXRα transcription. Although the LBD heterodimer dissociation mechanism is atypical, several studies have reported that pharmacological NR ligands can influence dimer formation in addition to their classical pharmacological properties in regulating coregulator protein interaction (*Kilu et al., 2021*; *Powell and Xu, 2008*; *Rehó et al., 2020*; *Tamrazi et al., 2002*). Our Nurr1-RXRα LBD heterodimer dissociation findings add to the repertoire of functional ligand targeting mechanisms of NRs that include classical pharmacological ligands, ligand-modulated posttranslational modification (*Choi et al., 2011*), PROTAC degrader molecules (*Flanagan and Neklesa, 2019*), and ligand-modulated NR phase separation/biomolecular condensates that include covalent targeting of the disordered N-terminal AF-1 domain (*Basu et al., 2022*; *Xie et al., 2022*).

Our data in support of a Nurr1-RXRα LBD heterodimer dissociation activation mechanism along with other published studies inform a model for activation of Nurr1-RXRα transcription by RXRα ligands (*Figure 8*). Nurr1 displays high transcriptional activity on monomeric NBRE sites as a monomer, and RXRα heterodimerization represses Nurr1 activity (*Aarnisalo et al., 2002*; *Forman et al., 1995*). RXRα does not bind to NBRE sequences but does interact with Nurr1 bound to NBRE DNA sequences via a protein-protein tethering interaction (*Sacchetti et al., 2002*) and recruits corepressor proteins that are released upon binding RXRα ligand (*Lammi et al., 2008*). Our RXRα truncation studies indicate that RXRα represses Nurr1 transcription through an RXRα LBD-dependent mechanism, pinpointing the Nurr1 and RXRα LBDs as the likely mode of tethering. Pharmacological agonists of RXRα, which stabilize an active RXRα LBD conformation (*De Lera et al., 2007*), as well as Nurr1-RXRα selective compounds (*Morita et al., 2005*; *Spathis et al., 2017*) that function as pharmacological antagonists of RXRα, dissociate tethered Nurr1-RXRα heterodimers leaving a monomeric Nurr1 bound to monomeric NBRE sites that has higher transcriptional activity than Nurr1-RXRα.

In contrast to the ligand-induced heterodimer dissociation activation mechanism, ligands that stabilize/strengthen Nurr1-RXRα heterodimerization or do not significantly affect Nurr1-RXRα dissociation—such as the RXRα antagonists in our study—may show more complex regulatory mechanisms. In addition to influencing Nurr1-RXRα interaction, antagonist ligands bound to RXRα, which is itself tethered to Nurr1 on NBRE genomic elements, could influence transcription of NBRE-driven gene transcription via classical pharmacological modulation mechanisms. RXRα-bound ligands will influence the conformation of and coregulator protein binding affinity to the RXRα AF-2 surface, which will in turn regulate recruitment of coregulator (coactivator or corepressor) proteins via interaction with the RXRα AF-2 surface.

In summary, our study shows that RXRα ligands can function as allosteric PPI inhibitors that bind to the canonical orthosteric ligand-binding pocket within the RXRα LBD and influence and Nurr1-mediated transcription via Nurr1-RXRα heterodimer dissociation. Our findings provide a molecular

understanding into the mechanism of action of Nurr1-RXRα activation that may influence the design and development of new compounds with therapeutic efficacy in PD, AD, and other aging-associated neurodegenerative disorders.

## Materials and methods

### Ligands, plasmids, and other reagents

All ligands were obtained from commercial vendors including Cayman Chemicals, Sigma, Axon Medchem, MedChemExpress, or Tocris Bioscience; BRF110 (CAS 2095489-35-1), HX600 (CAS 172705-89-4), 9cRA (CAS 5300-03-8), Bexarotene (CAS 153559-49-0), LG100268 (CAS 153559-76-3), CD3254 (CAS 196961-43-0), SR11237 (CAS 146670-40-8), UVI3003 (CAS 847239-17-2), LG100754 (CAS 180713-37-5), IRX4204 (CAS 220619-73-8), Rhein (CAS 478-43-3), HX531 (CAS 188844-34-0), Danthron (CAS 117-10-2), and PA452 (CAS 457657-34-0). FITC-labeled LXXLL-containing peptide derived from human PGC-1α (137-155; EAEEPSLLKKLLLAPANTQ) was synthesized by LifeTein with an N-terminal FITC label and an amidated C-terminus for stability. Bacterial expression plasmids included human Nurr1 (NR4A2) LBD (residues 353–598) (*de Vera et al., 2016*) and human RXRα (NR2B1) LBD (residues 223–462) (*Kojetin et al., 2015*) that were inserted into a pET45b(+) plasmid (Novagen) as a TEV-cleavable N-terminal hexahistidine(6xHis)-tag fusion protein. Luciferase reporter plasmids included a 3xNBRE-luciferase plasmid containing three copies of the NGFI-B response element corresponding to the monomeric binding site for Nurr1 *de Vera et al., 2016*; *Wilson et al., 1991*; and a 3xDR1-luciferase containing three copies of the optimal direct repeat 1 (DR1) binding site for RXRα (*Hughes et al., 2014*; *Subauste et al., 1994*). Mammalian expression plasmids included full-length human Nurr1 (residues 1–598) in pcDNA3.1 plasmid (*de Vera et al., 2016*) and full-length human RXRα (residues 1–462) in pCMV-Sport6 plasmid (*Zhang et al., 2011c*). To clone the RXRα ΔLBD construct (residues 1–226), site directed mutagenesis and PCR was used to insert a stop codon before the start of the LBD using the full-length RXRα expression plasmid the following primers: forward primer, CAGCAGCGCCTAAGAGGACATG; reverse primer, CATGTCCTCTTAGGCGCTGCTG. To clone the ΔNTD RXRα construct (residues 127–462), site directed mutagenesis and PCR was used to add a XhoI cut site and an ATG before at the end of the NTD using the following primers: forward primer, CCACCCCTCGAGAAACATGG; reverse primer, CCATGTTTCTCGAGGGGTGG—then restriction enzymes (XhoI and HindIII) were used to cut out the ΔNTD construct (DBD-hinge-LBD) for insertion using T4 ligase into pcDNA3.1 empty vector that had been linearized with XhoI and HindIII. To clone the RXRα LBD construct (200–462), XhoI was used to cut pcDNA3.1 and subsequently Gibson assembly was used to clone the LBD into the linearized vector using the gBlock sequence provided in *Supplementary file 1*.

### Cell lines for mammalian cell culture

All cells were obtained from and authenticated by ATCC and determined to be mycoplasma free. HEK293T (#CRL-11268) and SK-N-BE(2)-C (#CRL-2268) cells were cultured according to ATCC guidelines at low passage number (less than 10 passages; typically passages 2–4). HEK293T cells were grown at 37°C and 5% $CO_2$ in DEME (Gibco) supplemented with 10% fetal bovine serum (Gibco) and 100 units/mL of penicillin, 100 μg/mL of streptomycin, and 0.292 mg/mL of glutamine (Gibco) until 90–95% confluence in T-75 flasks prior to subculture or use. SK-N-BE(2)-C were grown at 37°C and 5% $CO_2$ in a media containing 1:1 mixture of EMEM (ATCC) and F12 medium (Gibco) supplemented with 10% fetal bovine serum (Gibco) until 90–95% confluence in T-75 flasks prior to subculture or use.

### Protein expression and purification

Proteins were expressed in *Escherichia coli* BL21(DE3) cells in autoinduction ZY media (without NMR isotopes) or M9 minimal media (using $^{15}NH_4Cl$ for NMR isotopic labeling). For autoinduction expression, cells were grown for 5 hr at 37°C, then 18 hr at 22°C, then centrifuged for harvesting. For M9 expression, cells were grown at 37°C and induced with 1.0 mM isopropyl β-D-thiogalactoside at OD (600 nm) of 0.6, grown for an additional 18 hr at 22°C, and then centrifuged for harvesting. Cell pellets were lysed using sonication and proteins were purified using Ni-NTA affinity chromatography and gel filtration/SEC. TEV protease was used to cleave the 6xHis-tag for all experiments except protein used for TR-FRET. The purified proteins were verified by SDS-PAGE, then stored in a buffer consisting of

20 mM potassium phosphate (pH 7.4), 50 mM potassium chloride, and 0.5 mM EDTA. All studies that used RXRα LBD protein used pooled SEC fractions for the apo-homodimeric form except for analytical SEC of the apo-tetrameric form.

## Transcriptional reporter assays

SK-N-BE(2)-C cells were seeded in 9.5 cm$^2$ cell culture well (Corning) at 0.5 million for transfection using Lipofectamine 2000 (Thermo Fisher Scientific) and Opti-MEM with an empty vector control or full-length Nurr1 expression plasmid (1 μg), with or without full-length or different RXRα truncation constructs (1 μg), and 3xNBRE-luc (1 μg). HEK293T cells were seeded in 9.5 cm$^2$ cell culture well (Corning) at 0.5 million for transfection using Lipofectamine 2000 (Thermo Fisher Scientific) and Opti-MEM with empty vector control or full-length RXRα expression plasmid (1 μg), and 3xDR1-luc (1 μg). After incubation for 16–20 hr, cells were transferred to white 384-well cell culture plates (Thermo Fisher Scientific) at 0.5 million cells/mL in 20 μL total volume/well. After a 4 hr incubation, cells were treated with 20 μL of vehicle control (DMSO) or 1 μM ligand. After a final 16–20 hr incubation, cells were harvested with 20 μL Britelite Plus (PerkinElmer), and luminescence was measured on a BioTek Synergy Neo multimode plate reader. The luminescence readouts were normalized to cells transfected with the empty vector (truncation construct assay) or cells treated with DMSO (assays with or without ligand treatment). Data were plotted using GraphPad Prism. In *Figure 1b*, statistical testing was performed and p-values were calculated in GraphPad Prism using the Brown-Forsythe and Welch multiple comparisons test (does not assume equal s.d. values among compared conditions) relative to full-length Nurr1 condition. In *Figures 3b and 4d*, statistical testing was performed and p-values were calculated in GraphPad Prism using the Brown-Forsythe and Welch multiple comparisons test (does not assume equal s.d. values among compared conditions) relative to DMSO control treated condition. Data are representative of two or more independent experiments (n=6 or 9 biological replicates).

## TR-FRET coactivator interaction assay

Assays were performed in 384-well black plates (Greiner) using 22.5 μL final well volume. Each well contained 4 nM 6xHis-RXRα LBD, 1 nM LanthaScreen Elite Tb-anti-His antibody (Thermo Fisher #PV5895), and 400 nM FITC-labeled PGC1α peptide in a buffer containing 20 mM potassium phosphate (pH 7.4), 50 mM KCl, 5 mM TCEP, and 0.005% Tween 20. Ligand stocks were prepared via serial dilution in DMSO and added to wells (10 μM final concentration) in triplicate. The plates were read using BioTek Synergy Neo multimode plate reader after incubation at 4°C for at least 2 hr. The Tb donor was excited at 340 nm, and its emission was measured at 495 nm; the emission of the acceptor FITC was measured at 520 nm. Data were plotted using GraphPad Prism as TR-FRET ratio (measurement at 520 nm/measurement at 495 nm) at a fixed ligand concentration (2–4 μM depending on the starting compound stock concentration). In *Figure 4b*, statistical testing was performed and p-values were calculated in GraphPad Prism using the Brown-Forsythe and Welch multiple comparisons test (does not assume equal s.d. values among compared conditions) relative to the DMSO control treated condition. Data are representative of two or more independent experiments (n=3 biological replicates) except for one RXRα ligand (IRX4204) where only one experiment (n=3 biological replicates) was performed.

## Isothermal titration calorimetry

Experiments were performed using a MicroCal iTC200. All experiments were solvent-matched and contained 0.25% DMSO (ligand vehicle) final concentration. RXRα LBD homodimerization affinity was measured using a dimer dissociation dilution method (*McPhail and Cooper, 1997*) by titrating 1 mM RXRα LBD from the syringe into the ITC buffer in the sample cell at 25°C with a 60 s delay between the 2 μL injections using a mixing speed of 1200 rpm for a total of 20 injections. Data were fitted using the dissociation model in MicroCal Origin 6.0 (MicroCal User Manual, section 7.6). RXRα LBD homodimer affinity ($K_D$) and enthalpy were determined to be 16.3 μM and –13.10 kcal/mol, respectively; this information was used in analysis of the interactions between Nurr1 LBD and ligand-bound RXRα LBD to approximate the dissociation step required for Nurr1 LBD to heterodimerize with RXRα LBD. ITC experiments of Nurr1 LBD to RXRα LBD (apo or ligand-bound) was performed by titrating Nurr1 LBD to RXRα LBD in a 10:1 ratio (500 or 1000 μM, and 50 or 100 μM, respectively) incubated with two equivalent of vehicle (DMSO) or ligand, and the run was performed using the same experimental

design. Experiments were performed at 25°C in duplicate, except for two RXRα ligands (IRX4204 and PA452) where only one experiment was performed; and one experimental replicate for HX600 was performed at 15°C, which gave a similar $K_D$ value to the data collected at 25°C. NITPIC software (*Keller et al., 2012*) was used to calculate baselines, integrate curves, prepare experimental data for fitting in SEDPHAT (*Brautigam et al., 2016*), which was used to obtain binding affinities and thermodynamic parameter measurements using a homodimer competition model (A + B + C <-> AB + C <-> AC + B; competing B and C for A, where A and C is RXRα LBD monomer and B is Nurr1 LBD monomer). Final figures were exported using GUSSI (*Brautigam, 2015*). In *Figure 5b*, statistical testing was performed and p-values were calculated in GraphPad Prism using ordinary one-way ANOVA relative to DMSO control treated condition.

## NMR spectroscopy

Two-dimensional [$^1$H,$^{15}$N]-TROSY-HSQC NMR experiments were performed at 298 K on a Bruker 700 MHz NMR instrument equipped with a QCI cryoprobe. Samples were prepared in a buffer containing 20 mM potassium phosphate (pH 7.4), 50 mM potassium chloride, 0.5 mM EDTA, and 10% D$_2$O. Experiments were collected using 200 µM $^{15}$N-labeled Nurr1 LBD with or without two molar equivalents of unlabeled RXRα LBD in the absence or presence of RXRα ligands. Data were processed and analyzed using NMRFx (*Norris et al., 2016*) using published NMR chemical shift assignments for the Nurr1 LBD (*Michiels et al., 2010*) that we validated and used previously (*de Vera et al., 2019*; *Munoz-Tello et al., 2020*). Relative Nurr1 LBD monomer populations were estimated by the relative peak intensities of the monomeric (I$_{monomer}$) and heterodimer (I$_{heterodimer}$) species using the following equation:

$$I_{monomer\_population} = \frac{I_{monomer}}{I_{monomer} + I_{heterodimer}}$$

NMR peak intensity errors for the monomeric (E$_{monomer}$) and heterodimeric (E$_{heterodimer}$) states were calculated in NMRFx and propagated in the I$_{monomer\_population}$ calculation using the following equation:

$$E_{monomer\_population} = \sqrt{\left(\frac{I_{monomer}}{(I_{heterodimer} + I_{monomer})2}\right) + \left(-\frac{I_{heterodimer}}{(I_{monomer} + I_{heterodimer})2}\right)}$$

Two replicates per condition were performed, except for two RXRα ligands (IRX4204 and PA452) where only one experiment was performed; average and s.d. calculations of the replicate I$_{monomer\_population}$ and E$_{monomer\_population}$ were performed in GraphPad Prism. In *Figure 6c*, statistical testing was performed and p-values were calculated in GraphPad Prism using ordinary one-way ANOVA relative to apo/DMSO control treated condition. We distinguished the two replicate measurements in the analysis, which were performed using different protein and ligand stock solutions, because one batch of samples (replicate 1) showed a more significant correlation to transcription than the other batch of samples.

## NMR lineshape simulations

NMR lineshape analysis was performed using NMR LineShapeKin version 4 (*Kovrigin, 2012*) and MATLAB R2022a via NMRbox (*Maciejewski et al., 2017*). 1D NMR lineshapes for Nurr1 LBD residue T411 ($^1$H$_N$ dimension), with or without two molar equivalents of RXRα LBD, were simulated using the U_L2 model (ligand binding to a receptor competing with ligand dimerization) where the ligand (RXRα LBD) exists as a homodimer that is dissociated into monomers upon binding to the receptor ($^{15}$N-labeled Nurr1 LBD). Simulations were performed using several experimentally defined parameters: ITC-determined apo-RXRα LBD homodimer affinity (16 µM); difference in chemical shift in hertz (150 Hz) between the monomer and heterodimer NMR peaks for T411 measured in 2D [$^1$H,$^{15}$N]-TROSY HSQC NMR data (w0 for R = –150 Hz; w0 for RL = 0 Hz); a receptor concentration of 200 µM (Rtotal = 2e-3); and ligand added at ±2 molar equivalents (LRratio = 1e-3 and 2.0); and relative relaxation rates for the monomer and heterodimer estimated to give peak full width half height (FWHH) line widths in 2D [$^1$H,$^{15}$N]-TROSY HSQC NMR data (for R, FWHH = 20, R2=10; for RL, FWHH = 40, R2=20 Hz); otherwise default parameters were used.

## Analytical SEC

To prepare Nurr1-RXRα LBD heterodimer for analytical SEC analysis, purified Nurr1 LBD and RXRα LBD (each 700 µM in 2.5 mL) were incubated together at 4°C in their storage buffer containing 20 mM potassium phosphate (pH 7.4), 50 mM potassium chloride, and 0.5 mM EDTA for 16 hr, injected (5 mL total) into HiLoad 16/600 Superdex 75 pg (Cytiva) connected to an AKTA FPLC system (GE Healthcare Life Science). Gel filtration was performed using the same buffer to purify Nurr1-RXRα LBD heterodimer for analysis, collecting 2 mL fraction to isolate heterodimer population into 15 samples of 0.5 mL (at a protein concentration of 300 µM) for analytical gel filtration. Additionally, purified Nurr1 LBD and RXRα LBD (pooled gel filtration fractions consisting of the homodimeric and homotetrameric species) were also used. Samples containing RXRα LBD (homodimer or heterodimer with Nurr1 LBD) were incubated with two molar equivalents of each ligand for 16 hr at 4°C before injection onto Superdex 75 Increase 10/300 GL (Cytiva) connected to the same AKTA FPLC system. The UV chromatograms were exported in CSV format and plotted using an in-house Jupyter notebook Python script using matplotlib and seaborn packages.

## Correlation and other statistical analyses

Correlation plots were performed using GraphPad Prism to calculate Pearson ($r_p$) and Spearman ($r_s$) correlation coefficients and two-tailed p-values between two experimental measurements per plot. Statistical testing of control to variable conditions was performed using one-way ANOVA testing as detailed above. PCA was performed in GraphPad Prism; all experimentally determined data were used as variables (method = standardize, PCs selected based on parallel analysis at the 95% percentile level with 1000 simulations). p-Value statistical significance shorthand conforms to GraphPad Prism standards: n.s., p>0.05; *, p≤0.05; **, p≤0.01; ***, p≤0.001; and ****, p≤0.0001.

## Acknowledgements

This work was supported by National Institutes of Health (NIH) grant R01AG070719 from the National Institute on Aging (NIA).

## Additional information

### Funding

| Funder | Grant reference number | Author |
| --- | --- | --- |
| National Institute on Aging | R01AG070719 | Douglas J Kojetin |

The funders had no role in study design, data collection and interpretation, or the decision to submit the work for publication.

### Author contributions

Xiaoyu Yu, Data curation, Formal analysis, Validation, Investigation, Visualization, Methodology, Writing – original draft, Writing – review and editing; Jinsai Shang, Data curation, Formal analysis, Validation, Investigation, Writing – review and editing; Douglas J Kojetin, Conceptualization, Formal analysis, Supervision, Funding acquisition, Validation, Investigation, Visualization, Methodology, Writing – original draft, Project administration, Writing – review and editing

### Author ORCIDs

Xiaoyu Yu ⓘ http://orcid.org/0000-0003-0549-9560
Douglas J Kojetin ⓘ http://orcid.org/0000-0001-8058-6168

### Decision letter and Author response

Decision letter https://doi.org/10.7554/eLife.85039.sa1
Author response https://doi.org/10.7554/eLife.85039.sa2

## Additional files

### Supplementary files

Supplementary file 1. gBlock sequence used to clone the retinoid X receptor alpha (RXRα) ligand-binding domain (LBD) only construct.

MDAR checklist

### Data availability

Raw ITC thermograms and fitted data are provided as Figure 5-source data 1. Input files for NMR LineShapeKin simulated NMR data analysis in MATLAB are provided as Figure 6-source code 1 (zip file including two input files and one readme file). Raw data used for graphical plots are provided as Figure 1-source data 1 (Nurr1 + RXRα truncated construct luciferase reporter data), Figure 3-source data 1 (RXRα ligand treated Nurr1-RXRα/3xNBRE3-luciferase reporter data), Figure 4-source data 1 (RXRα ligand treated RXRα LBD TR-FRET), Figure 4-source data 2 (RXRα ligand treated RXRα/3xDR1-luciferase reporter data), and Figure 6-source data 1 (RXRα ligand treated Nurr1-RXRα LBD NMR-observed monomer species). All other data generated or analyzed during this study are included in the manuscript and supporting files.

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

# Appendix 1

**Appendix 1—key resources table**

| Reagent type (species) or resource | Designation | Source or reference | Identifiers | Additional information |
|---|---|---|---|---|
| Strain, strain background (*Escherichia coli*) | BL21(DE3) | Sigma-Aldrich | CMC0014 | Electrocompetent cells |
| Cell line (*Homo sapiens*) | Human embryonic kidney epithelial | ATCC | CRL-11268 | |
| Cell line (*Homo sapiens*) | SK-N-BE(2) neuroblastoma | ATCC | CRL-2271 | |
| Chemical compound, drug | BRF110 | Sigma-Aldrich | CAS 2095489-35-1 | |
| Chemical compound, drug | HX600 | Axon Medchem | CAS 172705-89-4 | |
| Chemical compound, drug | 9-*cis*-Retinoic acid | Cayman Chemicals | CAS 5300-03-8 | |
| Chemical compound, drug | Bexarotene | Cayman Chemicals | CAS 153559-49-0 | |
| Chemical compound, drug | LG100268 | Cayman Chemicals | CAS 153559-76-3 | |
| Chemical compound, drug | CD3254 | Cayman Chemicals | CAS 196961-43-0 | |
| Chemical compound, drug | SR11237 | Tocris Bioscience | CAS 146670-40-8 | |
| Chemical compound, drug | UVI3003 | Cayman Chemicals | CAS 847239-17-2 | |
| Chemical compound, drug | LG100754 | Cayman Chemicals | CAS 180713-37-5 | |
| Chemical compound, drug | IRX4204 | MedChemExpress | CAS 220619-73-8 | |
| Chemical compound, drug | Rhein | Sigma-Aldrich | CAS 478-43- 3 | |
| Chemical compound, drug | HX531 | Cayman Chemicals | CAS 188844-34-0 | |
| Chemical compound, drug | Danthron | Sigma-Aldrich | CAS 117-10-2 | |
| Chemical compound, drug | PA452 | Tocris Bioscience | CAS 457657-34-0 | |
| Peptide, recombinant protein | FITC-PGC1α | LifeTein | | Amino acid sequence: EAEEPSLLKKLLLAPANTQ, with an N-terminal FITC label and an amidated C-terminus. |
| Recombinant DNA reagent | Nurr1-ligand binding domain (LBD) (plasmid) | *de Vera et al., 2016* | Bacteria expression plasmid | |
| Recombinant DNA reagent | RXRα-ligand binding domain (LBD) (plasmid) | *Kojetin et al., 2015* | Bacteria expression plasmid | |

*Appendix 1 Continued on next page*

*Appendix 1 Continued*

| Reagent type (species) or resource | Designation | Source or reference | Identifiers | Additional information |
|---|---|---|---|---|
| Recombinant DNA reagent | pET45b(+) | Novagen | 71327-3 | |
| Transfected construct (*Photinus pyralis*) | 3xNBRE-luciferase plasmid | ***de Vera et al., 2016*** | Sanger sequenced | |
| Transfected construct (*Photinus pyralis*) | 3xDR1-luciferase plasmid | ***Hughes et al., 2014*** | Mammalian expression plasmid, Sanger sequenced | The 3xPPRE-luciferase reporter plasmid in the referenced paper was used in our study. |
| Transfected construct (human) | Full-length human Nurr1 | ***de Vera et al., 2016*** | Mammalian expression plasmid, Sanger sequenced | |
| Transfected construct (human) | Full-length human RXRα in pCMV-Sport6 vector | ***Zhang et al., 2011c*** | Mammalian expression plasmid, Sanger sequenced | We obtained this plasmid from Griffin lab at UF Scripps Institute (see referenced paper). |
| Recombinant DNA reagent | pcDNA3.1 empty vector | Thermo Fisher Scientific | V790-20 | |
| Sequence-based reagent | RXRα-ΔLBD-F | This paper | PCR primer ordered from Sigma | CAGCAGCGCCTAAGAGGACATG |
| Sequence-based reagent | RXRα-ΔLBD-R | This paper | PCR primer ordered from Sigma | CATGTCCTCTTAGGCGCTGCTG |
| Sequence-based reagent | ΔNTD-RXRα-F | This paper | PCR primer ordered from Sigma | CCACCCCTCGAGAAACATGG |
| Sequence-based reagent | ΔNTD-RXRα-R | This paper | PCR primer ordered from Sigma | CCATGTTTCTCGAGGGGTGG |
| Gene (human) | Nurr1 (NR4A2) | Uniprot | | Full length: residues 1–598; LBD: residues 353–598 |
| Gene (human) | RXRα (NR2B1) | Uniprot | | Full length: residues 1–462; LBD: 223–462 |
| Sequence-based reagent | Restriction enzymes, ligase for cloning | NEB | | XhoI, HindIII, T4 ligase |
| Commercial assay or kit | Gibson assembly | NBE | E2611L | |
| Commercial assay or kit | Britelite plus Reporter Gene Assay System | Perkin Elmer | 6066769 | |

*Appendix 1 Continued on next page*

*Appendix 1 Continued*

| Reagent type (species) or resource | Designation | Source or reference | Identifiers | Additional information |
|---|---|---|---|---|
| Sequence-based reagent | RXRα-LBD | This paper | gBlock for Gibson assembly | CCAGCACAGTGGCGGCCGCA TGAAGCGGGAAGCCGTGCAG GAGGAGCGGCAGCGTGGCAA GGACCGGAACGAGAATGAGG TGGAGTCGACCAGCAGCGCC AACGAGGACATGCCGGTGGA GAGGATCCTGGAGGCTGAGC TGGCCGTGGAGCCCAAGACC GAGACCTACGTGGAGGCAAA CATGGGGCTGAACCCCAGCT CGCCGAACGACCCTGTCACC AACATTTGCCAAGCAGCCGA CAAACAGCTTTTCACCCTGG TGGAGTGGGCCAAGCGGATC CCACACTTCTCAGAGCTGCC CCTGGACGACCAGGTCATCC TGCTGCGGGCAGGCTGGAAT GAGCTGCTCATCGCCTCCTT CTCCCACCGCTCCATCGCCG TGAAGGACGGGATCCTCCTG GCCACCGGGCTGCACGTCCA CCGGAACAGCGCCCACAGCG CAGGGGTGGGCGCCATCTTT GACAGGGTGCTGACGGAGCT TGTGTCCAAGATGCGGGACA TGCAGATGGACAAGACGGAG CTGGGCTGCCTGCGCGCCAT CGTCCTCTTTAACCCTGACT CCAAGGGGCTCTCGAACCCG GCCGAGGTGGAGGCGCTGAG GGAGAAGGTCTATGCGTCCT TGGAGGCCTACTGCAAGCAC AAGTACCCAGAGCAGCCGGG AAGGTTCGCTAAGCTCTTGC TCCGCCTGCCGGCTCTGCGC TCCATCGGGCTCAAATGCCT GGAACATCTCTTCTTCTTCA AGCTCATCGGGGACACACCC ATTGACACCTTCCTTATGGA GATGCTGGAGGCGCCGCACC AAATGACTTGATCGAGTCTA GAGGGCCCG |
| Commercial assay or kit | LanthaScreen Elite Tb-anti-His antibody | Thermo Fisher | #PV5895 | |
| Software, algorithm | NITPIC software | *Keller et al., 2012* | | Baseline calculation, curve integration |
| Software, algorithm | SEDPHAT | *Brautigam et al., 2016* | | Estimation of binding affinity and thermodynamic parameter measurements |
| Software, algorithm | GUSSI | *Brautigam, 2015* | | Plot ITC figures |
| Other | NMR chemical shift assignment of Nurr1 LBD | *Michiels et al., 2010*; *de Vera et al., 2019*; *Munoz-Tello et al., 2020* | BMRB16541 | Published NMR peak assignment from Biological Magnetic Resonance Data Bank |
| Software, algorithm | NMRFx | *Norris et al., 2016* | | NMR data process and analysis |
| Software, algorithm | NMR LineShapeKin version 4 | *Kovrigin, 2012* | | NMR lineshape analysis |

*Appendix 1 Continued on next page*

*Appendix 1 Continued*

| Reagent type (species) or resource | Designation | Source or reference | Identifiers | Additional information |
|---|---|---|---|---|
| Software, algorithm | MATLAB R2022a via NMRbox | *Maciejewski et al., 2017* | | NMR lineshape analysis |
| Software, algorithm | Pearson and Spearman correlation analysis | GraphPad Prism | | Correlation analysis |
| Software, algorithm | Principal component analysis (PCA); | GraphPad Prism | | Correlation analysis |
| Software, algorithm | ANOVA multiple comparison test | GraphPad Prism | | Statistical testing |

