## [Editor Report]

This is a fundamental study of the activation process of Nurr1, an orphan nuclear receptor that may be a significant target for the treatment of neurodegenerative disorders. Nurr1 functions as a monomer, but may also heterodimerize with RXRα which represses Nurr1 transcriptional activation. The authors provide compelling evidence for Nurr1 activation through ligand-induced dissociation of an inactive Nurr1-RXRα heterodimer. These data will be important for biochemists and cell biologists working on regulatory / activation mechanisms of nuclear hormone receptors.

---

## [Decision Letter]

**Decision letter after peer review:**

Thank you for submitting your article "Molecular basis of ligand-dependent Nurr1-RXRα activation" for consideration by *eLife*. Your article has been reviewed by 3 peer reviewers, and the evaluation has been overseen by a Reviewing Editor and Volker Dötsch as the Senior Editor. The following individual involved in the review of your submission has agreed to reveal their identity: Sean W Fanning (Reviewer #2).

Essential revisions:

1) The RXRɑ antagonist HX531 has little to no effect on Nurr1-RXRɑ reporter activity like the other RXRɑ antagonists Rhein and Danthron (Figure 3b) but appears to selectively induce a >5-fold enhancement in the apparent affinity of LBD heterodimerization (Figure 5b) that seems inconsistent with the proposed model. The authors address a potential caveat with their analysis of the ITC data on lines 68-77 of page 4 but don't comment on any specific ligands or data. As a result, the correlation analyses presented in Figure 5c-d, although largely compelling, may be oversimplified. This is also true in Figure 6, where the correlation between reporter activity and Nurr1 monomerization assessed by NMR (Figure 6c) is much weaker than that with the ITC data (Figure 6e). Can the authors comment on the overall rigor and suitability of this simple correlative approach to justify these apparent discrepancies?

2) Related: In the correlation studies the authors include danthron and rhein as compounds in the Nurr1/RXR assay but not in the RXR PPI and RXR CTF assay. It is not clear if they are included in the correlation graphs in Figure 4 either; however, if they are excluded there should be a reason indicated. How do these compounds behave in the other assays?

3) Can the authors clarify how many independent times ITC data were collected for each ligand? The legend for Figure 5 states n = 1, but the methods state that at least two independent experiments were done. If so, the KD values in Figures 5 and 6 should reflect all of the data unless this is justified explicitly.

4) Related: Asterisks are used to indicate significance but the p-values assigned to them are not discussed in the figure legends nor in the methods. Also, in Figure 5b, error bars are not shown for the Kd values, and significance isn't shown between treatments. Adding in this statistical analysis would help judge whether dimerization differences are meaningful between ligands.

5) Since ITC is being used, information on changes to the heats, Gibbs free energy, enthalpy, and entropy should be available. Some discussion on whether changes to the observed binding affinities are ethically or entropically driven would be of interest to the biothermodynamics community.

6) Is there a way to estimate the error of measuring Nurr1 monomer populations by NMR in Figure 6a, b, c, and e? This could also be useful in making correlations in Figure 6.

7) Response at 1 μM was shown but no dose-response studies were performed to show differences in potency. These could be potentially correlated with the later studies to see if there's a relationship between transcriptional activities and dimerization affinity. Are such dose-response curves available?

8) Binding studies were only done for RXRalpha and the coregulator peptide. It would have been interesting to learn if the coregulator peptide binds the RXR-Nurr1 heterodimer. Comments?

9) HX531 appears to significantly enhance heterodimerization, is there any speculation as to why this is?

10) Should state in the cell culture section whether STR profiling was done to confirm cell line identity during the study, similar to mycoplasma testing.

11) In Figure 1, one of the key constructs (the δ LBD) shows high variability in its activity. Although it is not statistically different than the FL-Nurr1 control the variability that ranges from the FL-Nurr1 control to the FL-Nurr1/Fl-RARa group. Clearly, the LBD is playing a role, but my concern is that it might not be quite so clean. Is it possible that there is a contribution from the RXR DBD in this assay?

12) On page 4 line 43 the authors state,"…the two selective Nurr1-RXRa activating compounds BRF110 and HX600 function as pharmacological RXRa antagonists in these assays – a clear example that pharmacological RXRa modulation of Nurr-1/RXRa modulation is distinct". It isn't clear what assays the authors are referring to where these two compounds behaved as antagonists. This paragraph is prior to the presentation of the heterodimer assay where heterodimer formation is inhibited and in the previously presented assays these compounds were either activators (Nurr1/RXR), inactive (RXR coactivator PPI), or inactive/weak activators (RXR). Please clarify.

*Reviewer #1 (Recommendations for the authors):*

The RXRɑ antagonist HX531 has little to no effect on Nurr1-RXRɑ reporter activity like the other RXRɑ antagonists Rhein and Danthron (Figure 3b) but appears to selectively induce a >5-fold enhancement in the apparent affinity of LBD heterodimerization (Figure 5b) that seems inconsistent with the proposed model. The authors address a potential caveat with their analysis of the ITC data on lines 68-77 of page 4 but don't comment on any specific ligands or data. As a result, the correlation analyses presented in Figure 5c-d, although largely compelling, may be oversimplified. This is also true in Figure 6, where the correlation between reporter activity and Nurr1 monomerization assessed by NMR (Figure 6c) is much weaker than that with the ITC data (Figure 6e). Can the authors comment on the overall rigor and suitability of this simple correlative approach to justify these apparent discrepancies? I don't think more work necessarily needs to be done, but perhaps a more explicit description of caveats and possible implications for their model would be helpful.

Can the authors clarify how many independent times ITC data were collected for each ligand? The legend for Figure 5 states n = 1, but the methods state that at least two independent experiments were done. If so, the KD values in Figures 5 and 6 should reflect all of the data unless this is justified explicitly.

Is there a way to estimate the error of measuring Nurr1 monomer populations by NMR in Figure 6a, b, c, and e? This could also be useful in making correlations in Figure 6.

*Reviewer #2 (Recommendations for the authors):*

Congratulations on a compelling and high-impact study. I have a few recommendations that I believe will enhance the significance of this study.

Introduction

There's a typo in the first sentence of paragraph 2.

* are used for significance but the p-values assigned to them are not discussed in the figure legends nor in the methods that I could find.

Results

The second section discussing RXR ligands and Nurr1 transcriptional activity.

– Response at 1 μM was shown but no dose-response studies were performed to show differences in potency. These could be potentially correlated with the later studies to see if there's a relationship between transcriptional activities and dimerization affinity.

Third section looking at RXR coregulator binding.

– Binding studies were only done for RXRalpha and the coregulator peptide. It would have been interesting to learn if the coregulator peptide binds the RXR-Nurr1 heterodimer.

Fourth section using ITC to study heterodimerization.

In figure 5b, error bars are not shown for the Kd values, and significance isn't shown between treatments. Adding in this statistical analysis would help judge whether dimerization differences are meaningful between ligands.

– HX531 appears to significantly enhance heterodimerization, is there any speculation as to why this is?

– Since ITC is being used, you should have information on changes to the heats, Gibbs free energy, enthalpy, and entropy. Some discussion on whether changes to the observed binding affinities are enthalpically or entropically driven would be of interest to the biothermodynamics community.

Materials and methods

Should state in the cell culture section whether STR profiling was done to confirm cell line identity during your study, similar with mycoplasma testing.

Overall: Are there any inferences for this allostery that could be made based on existing RXRalpha co-crystal structures? An analysis of RXR allostery at the end could point to the mechanism of Nurr1 specificity. This would be a very interesting future study.

*Reviewer #3 (Recommendations for the authors):*

In Figure 1, one of the key constructs (the δ LBD) shows high variability in its activity. Although it is not statistically different than the FL-Nurr1 control the variability that ranges from the FL-Nurr1 control to the FL-Nurr1/Fl-RARa group. Clearly, the LBD is playing a role, but my concern is that it might not be quite so clean. Is it possible that there is a contribution from the RXR DBD in this assay? Otherwise, what do the authors believe causes this variability in this particular group?

In the correlation studies the authors include danthron and rhein as compounds in the Nurr1/RXR assay but not in the RXR PPI and RXR CTF assay. It is not clear if they are included in the correlation graphs in Figure 4 either; however, if they are excluded there should be a reason indicated. How do these compounds behave in the other assays?

On page 4 line 43 the authors state,"…the two selective Nurr1-RXRa activating compounds BRF110 and HX600 function as pharmacological RXRa antagonists in these assays – a clear example that pharmacological RXRa modulation of Nurr-1/RXRa modulation is distinct". It isn't clear to me what assays the authors are referring to where these two compounds behaved as antagonists. This paragraph is prior to the presentation of the heterodimer assay where heterodimer formation is inhibited and in the previously presented assays these compounds were either activators (Nurr1/RXR), inactive (RXR coactivator PPI), or inactive/weak activators (RXR).

---

## [Author Response]

Essential revisions:1) The RXRɑ antagonist HX531 has little to no effect on Nurr1-RXRɑ reporter activity like the other RXRɑ antagonists Rhein and Danthron (Figure 3b) but appears to selectively induce a >5-fold enhancement in the apparent affinity of LBD heterodimerization (Figure 5b) that seems inconsistent with the proposed model. The authors address a potential caveat with their analysis of the ITC data on lines 68-77 of page 4 but don't comment on any specific ligands or data. As a result, the correlation analyses presented in Figure 5c-d, although largely compelling, may be oversimplified. This is also true in Figure 6, where the correlation between reporter activity and Nurr1 monomerization assessed by NMR (Figure 6c) is much weaker than that with the ITC data (Figure 6e). Can the authors comment on the overall rigor and suitability of this simple correlative approach to justify these apparent discrepancies?

We do not believe that the RXRα antagonist data is inconsistent with the ligand-induced activation model and have additional text to the manuscript discussion and a new Results section titled “*PCA reveals data features associated with Nurr1-RXRα agonism*” where we used principal component analysis (PCA) of all of the experimental data and a new discussion paragraph (second to last) describing how Nurr1-RXRα agonism/heterodimer dissociation and antagonism/heterodimer stabilization likely function via different regulatory mechanisms.

The reviewer also questions the significance of the correlation analysis performed in our study. In our original submission, we used a linear correlation analysis by plotting two sets of experimentally determined data, fitting the data via linear regression, and reporting R^2^ values as a description of correlation. However, since linear fits are not appropriate in cases where both variables are experimentally measured, in this revised manuscript we instead performed Pearson (r_p_) and Spearman (r_s_) correlation analyses and report the associated p-values that report on whether the correlation coefficients (r_p_ or r_s_) are statistically significant (see grey boxes above plots). Furthermore, the new PCA analysis we added also enhances the rigor of this analysis by analyzing all of the experimental data together in an unbiased manner.

2) Related: In the correlation studies the authors include danthron and rhein as compounds in the Nurr1/RXR assay but not in the RXR PPI and RXR CTF assay. It is not clear if they are included in the correlation graphs in Figure 4 either; however, if they are excluded there should be a reason indicated. How do these compounds behave in the other assays?

We did not include danthron and rhein in the RXRα LBD coactivator peptide interaction TR-FRET assay (Figure 4b) because the compounds are colored and would interfere with the assay. We ran the RXRα/3xDR1-luciferase cellular transcriptional reporter assay (Figure 4d) at the same time as the TR-FRET assay, and therefore we did not include danthron and rhein since no comparison to the TR-FRET data could be made. Danthron and rhein data from the Nurr1-RXRα/3xNBRE-luciferase cellular transcriptional reporter assay are not included in the correlation plots in Figure 4e-g.

Danthron and rhein do not appear to cause interference in the 3xNBRE-luciferase cellular reporter assay we used to measure Nurr1-RXRα transcription, nor in the ITC, NMR, or SEC experiments that measure Nurr1-RXRα LBD interaction; and were therefore included in correlation plots among these experiments.

3) Can the authors clarify how many independent times ITC data were collected for each ligand? The legend for Figure 5 states n = 1, but the methods state that at least two independent experiments were done. If so, the KD values in Figures 5 and 6 should reflect all of the data unless this is justified explicitly.

In this revision the Figures now show data points for duplicate ITC measurements for all but two RXRα ligands (PA452 and IRX4204).

4) Related: Asterisks are used to indicate significance but the p-values assigned to them are not discussed in the figure legends nor in the methods. Also, in Figure 5b, error bars are not shown for the Kd values, and significance isn't shown between treatments. Adding in this statistical analysis would help judge whether dimerization differences are meaningful between ligands.

As requested by the reviewer, we performed statistical testing and p-value analysis for ligand-treated conditions to control conditions (apo/DMSO) for all experimentally measured values in all relevant figures and ensured that all plots include error bars.

5) Since ITC is being used, information on changes to the heats, Gibbs free energy, enthalpy, and entropy should be available. Some discussion on whether changes to the observed binding affinities are ethically or entropically driven would be of interest to the biothermodynamics community.

As requested by the reviewer, we included the measured and fitted ITC thermodynamic parameters to Table 1 (new in the revised manuscript), perform correlation analyses between binding affinity and enthalpy/entropy, and include a discussion of the data and analyses.

6) Is there a way to estimate the error of measuring Nurr1 monomer populations by NMR in Figure 6a, b, c, and e? This could also be useful in making correlations in Figure 6.

We updated Figure 6 to include peak intensity errors calculated by the software we used for NMR analysis; additional details can be found in the methods section.

7) Response at 1 μM was shown but no dose-response studies were performed to show differences in potency. These could be potentially correlated with the later studies to see if there's a relationship between transcriptional activities and dimerization affinity. Are such dose-response curves available?

In this study we are interested in the functional efficacy of the ligand when completely bound to RXRα and therefore performed studies using a single concentration of ligand that is near/above their reported RXRα LBD binding affinity. It is possible there could be a correlation between ligand potency and function, or heterodimerization with Nurr1 could influence ligand affinity/potency vs. RXRα homodimers, which could be apparent in ligand dose-response experiments; we plan to investigate this in future studies.

8) Binding studies were only done for RXRalpha and the coregulator peptide. It would have been interesting to learn if the coregulator peptide binds the RXR-Nurr1 heterodimer. Comments?

The Nurr1 LBD contains a reversed “charge clamp” in its AF-2 surface—a proper charge clamp is important for binding LXXLL-containing motifs present within transcriptional coactivator proteins—and as such Nurr1 does not interact with canonical nuclear receptor coregulator proteins. We updated the first paragraph of the Results section titled “*Nurr1-RXRα activation is not correlated with pharmacological RXRα agonism*” and included three references to introduce this concept.

If were to perform the TR-FRET coactivator peptide interaction assay to determine how RXRα ligands influence coactivator peptide interaction with Nurr1-RXRα, we anticipate the results would be confounded by several different mechanisms including ligand-dependent Nurr1-RXRα dissociation; ligand-dependent changes in coactivator peptide interaction with RXRα; and ligand-dependent changes in RXRα oligomerization as our size exclusion chromatography (SEC) data indicate ligands differentially affect the hydrodynamic solution properties of RXRα LBD.

9) HX531 appears to significantly enhance heterodimerization, is there any speculation as to why this is?

Our size exclusion chromatography (SEC) data (Figure 6—figure supplement 2) shows that HX531 decreases the hydrodynamic radius of the RXRα LBD (shifts the SEC peak to the right compared to apo-RXRα LBD), which could indicate HX531 changes the RXRα LBD monomer-dimer equilibrium towards a monomeric state (or more generally towards a homodimer conformation that is more compact). Because Nurr1 heterodimerization with RXRα competes for RXRα LBD homodimerization, ligands that favor a monomeric RXRα LBD state would result in higher affinity heterodimerization with Nurr1 LBD. This is an interesting point that we plan to investigate in future studies.

10) Should state in the cell culture section whether STR profiling was done to confirm cell line identity during the study, similar to mycoplasma testing.

We updated the methods cell culture section to state that cells were authenticated by ATCC and cultured at low passage number (less than 10 passages).

11) In Figure 1, one of the key constructs (the δ LBD) shows high variability in its activity. Although it is not statistically different than the FL-Nurr1 control the variability that ranges from the FL-Nurr1 control to the FL-Nurr1/Fl-RARa group. Clearly, the LBD is playing a role, but my concern is that it might not be quite so clean. Is it possible that there is a contribution from the RXR DBD in this assay?

In the first paragraph of the Results section, we updated the description of these data in the Results section and added p-values for the statistical comparison of the RXRα constructs (full-length and truncation) vs. full-length Nurr1. We also added additional information and two more references that implicate a Nurr1 LBD-RXRα LBD interaction in the mechanism by which RXRα represses Nurr1 transcription.

12) On page 4 line 43 the authors state,"…the two selective Nurr1-RXRa activating compounds BRF110 and HX600 function as pharmacological RXRa antagonists in these assays – a clear example that pharmacological RXRa modulation of Nurr-1/RXRa modulation is distinct". It isn't clear what assays the authors are referring to where these two compounds behaved as antagonists. This paragraph is prior to the presentation of the heterodimer assay where heterodimer formation is inhibited and in the previously presented assays these compounds were either activators (Nurr1/RXR), inactive (RXR coactivator PPI), or inactive/weak activators (RXR). Please clarify.

We revised this section of the manuscript removing this confusing statement and added a new section at the end of the Results section titled “*Nurr1-RXRα activation is not correlated with pharmacological RXRα agonism*” to more concisely make the point that ligand-dependent RXRα homodimer modulation and Nurr1-RXRα heterodimer modulation may function through distinct mechanisms.

Reviewer #1 (Recommendations for the authors):The RXRɑ antagonist HX531 has little to no effect on Nurr1-RXRɑ reporter activity like the other RXRɑ antagonists Rhein and Danthron (Figure 3b) but appears to selectively induce a >5-fold enhancement in the apparent affinity of LBD heterodimerization (Figure 5b) that seems inconsistent with the proposed model. The authors address a potential caveat with their analysis of the ITC data on lines 68-77 of page 4 but don't comment on any specific ligands or data. As a result, the correlation analyses presented in Figure 5c-d, although largely compelling, may be oversimplified. This is also true in Figure 6, where the correlation between reporter activity and Nurr1 monomerization assessed by NMR (Figure 6c) is much weaker than that with the ITC data (Figure 6e). Can the authors comment on the overall rigor and suitability of this simple correlative approach to justify these apparent discrepancies? I don't think more work necessarily needs to be done, but perhaps a more explicit description of caveats and possible implications for their model would be helpful.

Addressed in our response to Essential revisions #1.

Can the authors clarify how many independent times ITC data were collected for each ligand? The legend for Figure 5 states n = 1, but the methods state that at least two independent experiments were done. If so, the KD values in Figures 5 and 6 should reflect all of the data unless this is justified explicitly.

Addressed in our response to Essential revisions #3.

Is there a way to estimate the error of measuring Nurr1 monomer populations by NMR in Figure 6a, b, c, and e? This could also be useful in making correlations in Figure 6.

Addressed in our response to Essential revisions #6.

Reviewer #2 (Recommendations for the authors):Congratulations on a compelling and high-impact study. I have a few recommendations that I believe will enhance the significance of this study.IntroductionThere's a typo in the first sentence of paragraph 2.

We fixed it, thanks!

* are used for significance but the p-values assigned to them are not discussed in the figure legends nor in the methods that I could find.

We added the p-values to the figures and described the statistical testing in the figure legends and Materials and methods section.

ResultsThe second section discussing RXR ligands and Nurr1 transcriptional activity.– Response at 1 μM was shown but no dose-response studies were performed to show differences in potency. These could be potentially correlated with the later studies to see if there's a relationship between transcriptional activities and dimerization affinity.

Addressed in our response to Essential revisions #7.

Third section looking at RXR coregulator binding.– Binding studies were only done for RXRalpha and the coregulator peptide. It would have been interesting to learn if the coregulator peptide binds the RXR-Nurr1 heterodimer.

Addressed in our response to Essential revisions #8.

Fourth section using ITC to study heterodimerization.In figure 5b, error bars are not shown for the Kd values, and significance isn't shown between treatments. Adding in this statistical analysis would help judge whether dimerization differences are meaningful between ligands.

We used GraphPad Prism to calculate Pearson (r_p_) and Spearman (r_s_) correlation coefficients and two-tailed p-values to all appropriate figure panels.

– HX531 appears to significantly enhance heterodimerization, is there any speculation as to why this is?

Addressed in our response to Essential revisions #1.

– Since ITC is being used, you should have information on changes to the heats, Gibbs free energy, enthalpy, and entropy. Some discussion on whether changes to the observed binding affinities are enthalpically or entropically driven would be of interest to the biothermodynamics community.

Addressed in our response to Essential revisions #5.

Materials and methodsShould state in the cell culture section whether STR profiling was done to confirm cell line identity during your study, similar with mycoplasma testing.

Addressed in our response to Essential revisions #10.

Overall: Are there any inferences for this allostery that could be made based on existing RXRalpha co-crystal structures? An analysis of RXR allostery at the end could point to the mechanism of Nurr1 specificity. This would be a very interesting future study.

We added a new paragraph to the end of the Results section “*Nurr1-RXRα activation is correlated with dissociation of Nurr1 LBD monomer*” that describes how our data along with previously reported studies inform on the potential allosteric mechanism by which RXRα ligands influence Nurr1-RXRα heterodimer dissociation. We agree that further studies are warranted to explore this further.

Reviewer #3 (Recommendations for the authors):In Figure 1, one of the key constructs (the δ LBD) shows high variability in its activity. Although it is not statistically different than the FL-Nurr1 control the variability that ranges from the FL-Nurr1 control to the FL-Nurr1/Fl-RARa group. Clearly, the LBD is playing a role, but my concern is that it might not be quite so clean. Is it possible that there is a contribution from the RXR DBD in this assay? Otherwise, what do the authors believe causes this variability in this particular group?

Addressed in our response to Essential revisions #11.

In the correlation studies the authors include danthron and rhein as compounds in the Nurr1/RXR assay but not in the RXR PPI and RXR CTF assay. It is not clear if they are included in the correlation graphs in Figure 4 either; however, if they are excluded there should be a reason indicated. How do these compounds behave in the other assays?

Addressed in our response to Essential revisions #2.

On page 4 line 43 the authors state,"…the two selective Nurr1-RXRa activating compounds BRF110 and HX600 function as pharmacological RXRa antagonists in these assays – a clear example that pharmacological RXRa modulation of Nurr-1/RXRa modulation is distinct". It isn't clear to me what assays the authors are referring to where these two compounds behaved as antagonists. This paragraph is prior to the presentation of the heterodimer assay where heterodimer formation is inhibited and in the previously presented assays these compounds were either activators (Nurr1/RXR), inactive (RXR coactivator PPI), or inactive/weak activators (RXR).

Addressed in our response to Essential revisions #12.